# Inferring and validating mechanistic models of neural microcircuits based on spike-train data

Josef Ladenbauer [1]*, Sam McKenzie[2], Daniel Fine English [3], Olivier Hagens[4] & Srdjan Ostojic [1]*

The interpretation of neuronal spike train recordings often relies on abstract statistical models that allow for principled parameter estimation and model selection but provide only limited insights into underlying microcircuits. In contrast, mechanistic models are useful to interpret microcircuit dynamics, but are rarely quantitatively matched to experimental data due to methodological challenges. Here we present analytical methods to efficiently fit spiking circuit models to single-trial spike trains. Using derived likelihood functions, we statistically infer the mean and variance of hidden inputs, neuronal adaptation properties and connectivity for coupled integrate-and-fire neurons. Comprehensive evaluations on synthetic data, validations using ground truth in-vitro and in-vivo recordings, and comparisons with existing techniques demonstrate that parameter estimation is very accurate and efficient, even for highly subsampled networks. Our methods bridge statistical, data-driven and theoretical, model-based neurosciences at the level of spiking circuits, for the purpose of a quantitative, mechanistic interpretation of recorded neuronal population activity.

---

[1] Laboratoire de Neurosciences Cognitives et Computationnelles, INSERM U960, École Normale Supérieure, PSL Research University, 29 rue d'Ulm, 75005 Paris, France. [2] Neuroscience Institute, New York University, 20 Cooper Square, New York, NY 10003, USA. [3] School of Neuroscience, Virginia Tech, 970 Washington St. SW, Blacksburg, VA 24061, USA. [4] Laboratory of Neural Microcircuitry, Brain Mind Institute, School of Life Sciences, École Polytechnique Fédérale de Lausanne, 1015 Lausanne, Switzerland. *email: josef.ladenbauer@gmail.com; srdjan.ostojic@ens.fr

In recent years neuronal spike train data have been collected at an increasing pace, with the ultimate aim of unraveling how neural circuitry implements computations that underlie behavior. Often these data are acquired from extracellular electrophysiological recordings in vivo without knowledge of neuronal input and connections between neurons. To interpret such data, the recorded spike trains are frequently analyzed by fitting abstract parametric models that describe statistical dependencies in the data. A typical example consists in fitting generalized linear models (GLMs) to characterize the mapping between measured sensory input and neuronal spiking activity[1–5]. These approaches are very useful for quantifying the structure in the data, and benefit from statistically principled parameter estimation and model selection methods. However, their interpretive power is limited as the underlying models typically do not incorporate prior biophysical constraints.

Mechanistic models of coupled neurons on the other hand involve biophysically interpretable variables and parameters, and have proven essential for analyzing the dynamics of neural circuits. At the top level of complexity for this purpose are detailed models of the Hodgkin–Huxley type[6,7]. Comparisons between models of this type and recorded spike trains have revealed that multiple combinations of biophysical parameters give rise to the same firing patterns[8,9]. This observation motivates the use of models at an intermediate level of complexity, and in particular integrate-and-fire (I&F) neurons, which implement in a simplified manner the key biophysical constraints with a reduced number of effective parameters and can be equipped with various mechanisms, such as spike initiation[10–12], adaptive excitability[13,14], or distinct compartments[15,16] to generate diverse spiking behaviors[17,18] and model multiple neuron types[19,20]. I&F models can reproduce and predict neuronal activity with a remarkable degree of accuracy[11,21,22], essentially matching the performance of biophysically detailed models with many parameters[17,23]; thus, they have become state-of-the-art models for describing neural activity in in vivo-like conditions[11,19,20,24]. In particular, they have been applied in a multitude of studies on local circuits[25–31], network dynamics[32–36], learning/computing in networks[37–40], as well as in neuromorphic hardware systems[37,41–43].

I&F neurons can be fit in straightforward ways to membrane voltage recordings with knowledge of the neuronal input, typically from in vitro preparations[11,17,19,20,24,44,45]. Having only access to the spike times as in a typical in vivo setting however poses a substantial challenge for the estimation of parameters. Estimation methods that rely on numerical simulations to maximize a likelihood or minimize a cost function[46–48] strongly suffer from the presence of intrinsic variability in this case.

To date, model selection methods based on extracellular recordings are thus much more advanced and principled for statistical/phenomenological models than for mechanistic circuit models. To bridge this methodological gap, here we present analytical tools to efficiently fit I&F circuits to observed spike times from a single trial. By maximizing analytically computed likelihood functions, we infer the statistics of hidden inputs, input perturbations, neuronal adaptation properties and synaptic coupling strengths, and evaluate our approach extensively using synthetic data. Importantly, we validate our inference methods for all of these features using ground truth in vitro and in vivo data from whole-cell[49,50] and combined juxtacellular–extracellular recordings[51]. Systematic comparisons with existing model-based and model-free methods on synthetic data and electrophysiological recordings[49–52] reveal a number of advantages, in particular for the challenging task of estimating synaptic couplings from highly subsampled networks.

## Results

**Maximum likelihood estimation for I&F neurons.** Maximum likelihood estimation is a principled method for fitting statistical models to observations. Given observed data D and a model that depends on a vector of parameters $\boldsymbol{\theta}$, the estimated value of the parameter vector is determined by maximizing the likelihood that the observations are generated by the model, $p(D|\boldsymbol{\theta})$, with respect to $\boldsymbol{\theta}$. This method features several attractive properties, among them: (1) the distribution of maximum likelihood estimates is asymptotically Gaussian with mean given by the true value of $\boldsymbol{\theta}$; (2) the variances of the parameter estimates achieve a theoretical lower bound, the Cramer–Rao bound, as the sample size increases[53].

Let us first focus on single neurons. The data we have are neuronal spike times, which we collect in the ordered set $D := \{t_1,\ldots,t_K\}$. We consider neuron models of the I&F type, which describe the membrane voltage dynamics by a differential equation together with a reset condition that simplifies the complex, but rather stereotyped, dynamics of action potentials (spikes). Here, we focus on the classical leaky I&F model[54] but also consider a refined variant that includes a nonlinear description of the spike-generating sodium current at spike initiation and is known as exponential I&F model[10]. An extended I&F model that accounts for neuronal spike rate adaptation[14,17,55] is included in Results section "Inference of neuronal adaptation". Each model neuron receives fluctuating inputs described by a Gaussian white noise process with (time-varying) mean $\mu(t)$ and standard deviation $\sigma$ (for details on the models see Methods section "I&F neuron models").

We are interested in the likelihood $p(D|\boldsymbol{\theta})$ of observing the spike train D from the model with parameter vector $\boldsymbol{\theta}$. As spike emission in I&F models is a renewal process (except in presence of adaptation, see below) this likelihood can be factorized as

$$p(D|\boldsymbol{\theta}) = \prod_{k=1}^{K-1} p(t_{k+1}|t_k, \mu[t_k, t_{k+1}], \boldsymbol{\theta}), \qquad (1)$$

where $\mu[t_k, t_{k+1}] := \{\mu(t)|t \in [t_k, t_{k+1}]\}$ denotes the mean input time series across the time interval $[t_k, t_{k+1}]$. In words, each factor in Eq. (1) is the probability density value of a spike time conditioned on knowledge about the previous spike time, the parameters contained in $\boldsymbol{\theta}$ and the mean input time series across the inter-spike interval (ISI). Below we refer to these factors as conditioned spike time likelihoods. We assume that $\mu[t_k, t_{k+1}]$ can be determined using available knowledge, which includes the parameters in $\boldsymbol{\theta}$ as well as the observed spike times. Note that we indicate the parameter $\mu$ separately from $\boldsymbol{\theta}$ due to its exceptional property of variation over time.

For robust and rapid parameter estimation using established optimization techniques we need to compute $p(D|\boldsymbol{\theta})$ as accurately and efficiently as possible. Typical simulation-based techniques are not well suited because they can only achieve a noisy approximation of the likelihood that depends on the realization of the input fluctuations and is difficult to maximize. This poses a methodological challenge which can be overcome using analytical tools that have been developed for I&F neurons in the context of the forward problem of calculating model output for given parameters[56–60]. These tools led us to the following methods that we explored for the inverse problem of parameter estimation:

- Method 1 calculates the factors of the likelihood (Eq. (1), right hand side) by solving a Fokker–Planck PDE using suitable numerical solution schemes (for details see Methods section "Method 1: conditioned spike time likelihood"). In model scenarios where the mean input is expected to vary only little in $[t_k, t_{k+1}]$ and contains perturbations with weak effects, for

example, synaptic inputs that cause relatively small post-synaptic potentials, we can write $\mu(t) = \mu_0^k + J\mu_1(t)$ with small $|J|$, where $\mu_0^k$ may vary between ISIs. This allows us to employ the first order approximation

$$p(t_{k+1}|t_k, \mu[t_k, t_{k+1}], \boldsymbol{\theta}) \approx p_0(t_{k+1}|t_k, \boldsymbol{\theta}) + J\, p_1(t_{k+1}|t_k, \mu_1[t_k, t_{k+1}], \boldsymbol{\theta}),$$
(2)

where $\boldsymbol{\theta}$ contains parameters that remain constant within ISIs, including $\mu_0^k$; $J$ is indicated separately for improved clarity. $p_0$ denotes the conditioned spike time likelihood in absence of input perturbations and $p_1$ the first order correction. We indicate the use of this approximation explicitly by method 1a.

- Method 2 involves an approximation of the spike train by an inhomogeneous Poisson point process. The spike rate $r(t)$ of that process is effectively described by a simple differential equation derived from the I&F model and depends on the mean input up to time $t$ as well as the other parameters in $\boldsymbol{\theta}$ (for details see Methods section "Method 2: derived spike rate model"). In this case the factors in Eq. (1), right, are expressed as

$$p(t_{k+1}|t_k, \mu[t_k, t_{k+1}], \boldsymbol{\theta}) \approx$$
$$r(t_{k+1}|\mu[t_1, t_{k+1}], \boldsymbol{\theta})\, \exp\left(-\int_{t_k}^{t_{k+1}} r(\tau|\mu[t_1, \tau], \boldsymbol{\theta})d\tau\right).$$
(3)

For each of these methods the likelihood (Eq. (1)) is then maximized using well-established algorithms (see Methods section "Likelihood maximization"). Notably, for the leaky I&F model this likelihood is mathematically guaranteed to be free from local maxima[61] so that any optimization algorithm will converge to its global maximum, which ensures reliable and tractable fitting.

The most accurate and advantageous method depends on the specific setting, as illustrated for different scenarios in the following sections. Several scenarios further allow to benchmark our methods against a related approach from Pillow and colleagues, which also uses the Fokker–Planck equation[61,62]. We compared the different methods in terms of estimation accuracy and computation time. For information on implementation and computational demands see Methods section "Implementation and computational complexity".

**Inference of background inputs**. We first consider the spontaneous activity of an isolated recorded neuron. This situation is modeled with an I&F neuron receiving a stationary noisy background input with constant mean $\mu$ and standard deviation $\sigma$. For this scenario method 1 is more accurate than method 2 while similarly efficient, and therefore best suited. The parameters of interest for estimation are limited to $\mu$ and $\sigma$, together with the membrane time constant $\tau_m$ (for details see Methods section "I&F neuron models").

An example of the simulated ground truth data, which consists of the membrane voltage time series including spike times, is shown in Fig. 1a together with ISI and membrane voltage histograms. Note that for estimation we only use the spike times. By maximizing the likelihood the true parameter values are well recovered (Fig. 1b) and we obtain an accurate estimate of the ISI density. In addition, we also obtain an accurate estimate for the unobserved membrane voltage density, which can be calculated using a slight modification of method 1 in a straightforward way once the parameter values are determined. Interestingly, the fits are accurate regardless of whether the membrane time constant $\tau_m$ is set to the true value (Fig. 1a, b), estimated or set to a wrong

value within a biologically plausible range (Fig. 1c and Supplementary Fig. 1a).

We next evaluated the estimation accuracy for different numbers of observed spikes (Fig. 1d). As little as 50 spikes already lead to a good solution with a maximum average relative error of about 10%. Naturally, the estimation accuracy increases with the number of observed spikes. Moreover, the variance of the parameter estimates decreases as the number of spikes increases (see insets in Fig. 1d).

To further quantify how well the different parameters can be estimated from a spike train of a given length, we analytically computed the Cramer–Rao bound (see Methods section "Calculation of the Cramer–Rao bound"). This bound limits the variance of any unbiased estimator from below and is approached by the variance of a maximum likelihood estimator. For $\tau_m$ fixed and a reasonable range of values for $\mu$ and $\sigma$ we consistently find that the variance of the estimates of both input parameters decreases with decreasing ISI variability (Fig. 1e) and the variance for $\mu$ is smaller than that for $\sigma$ (Fig. 1d, e). If $\tau_m$ is included in the estimation, the variance of its estimates is by far the largest and that for $\sigma$ the smallest (in relation to the range of biologically plausible values, Supplementary Fig. 1b, c). Together with the results from Fig. 1c and Supplementary Fig. 1a this motivates to keep $\tau_m$ fixed.

Comparisons of our method with the approach from Pillow et al.[62] show clear improvements on estimation accuracy (Supplementary Fig. 1c) and computation time (Supplementary Fig. 1d: reduction by two orders of magnitude). Finally, we tested our method on the exponential I&F model, which involves an additional nonlinearity and includes a refractory period. For this model we also obtain accurate estimates (Supplementary Fig. 1e).

We validated our inference method using somatic whole-cell recordings of cortical pyramidal cells (PYRs)[49] and fast-spiking interneurons (INTs) exposed to injected fluctuating currents. A range of stimulus statistics, in terms of different values for the mean $\mu_I$ and standard deviation $\sigma_I$ of these noise currents, was applied and each cell responded to multiple different stimuli (examples are shown in Fig. 2a; for details see Methods section "In vitro ground truth data on neuronal input statistics"). We estimated the input parameters $\mu$ and $\sigma$ of an I&F neuron from the observed spike train for each stimulus by maximizing the spike train likelihood.

Model fitting yielded an accurate reproduction of the ISI distributions (Fig. 2a). Importantly, the estimated input statistics captured well the true stimulus statistics (Fig. 2b, c). In particular, estimated and true mean input as well as estimated and true input standard deviations were strongly correlated for all cells (Fig. 2b, c). The correlation coefficients between estimated and ground truth values for INTs are larger than those for PYRs, as reflected by the concave shape of the estimated $\mu$ values as a function of $\mu_I$. This shape indicates a saturation mechanism that is not included in the I&F model. Indeed, it can in part be explained by the intrinsic adaptation property of PYRs (see Results section "Inference of neuronal adaptation"). Furthermore, correlation coefficients are slightly increased for longer stimuli (15 s compared to 5 s duration) due to improved estimation accuracy for longer spike trains (Supplementary Fig. 2). Comparison with a Poisson point process showed that the I&F model is the preferred one across all cells and stimuli according to the Akaike information criterion (AIC), which takes into account both goodness of fit and complexity of a model (Fig. 2d).

Noise injections in vitro mimic in a simplified way the background inputs that lead to spontaneous neural activity in vivo, and certain dynamical aspects may not be well captured. Therefore, we additionally considered spike-train data obtained from extracellular multi-channel recordings in primary auditory

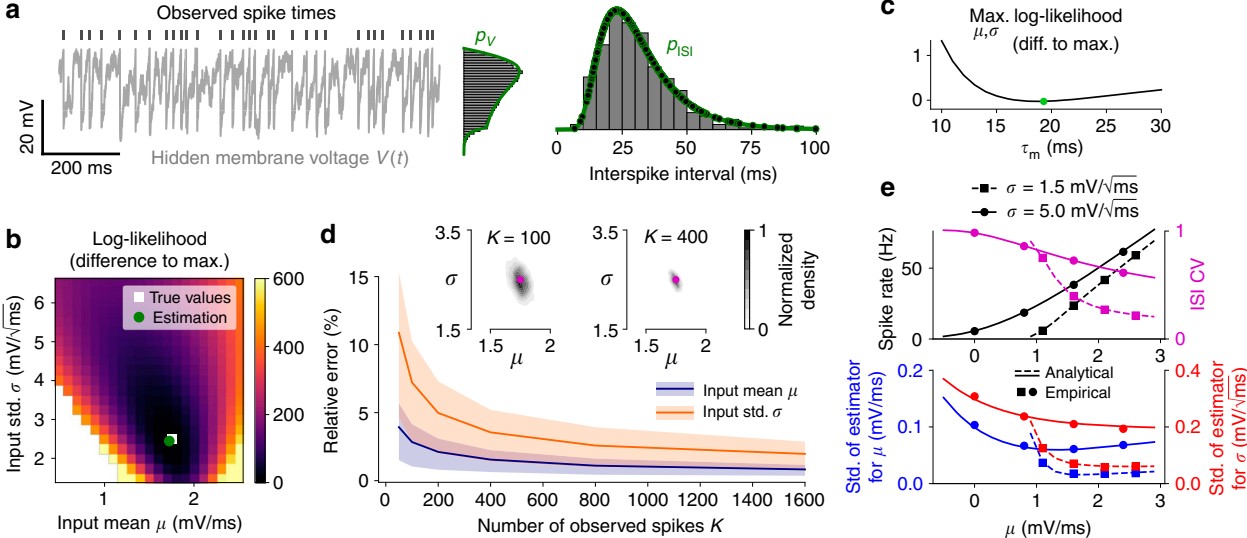

**Fig. 1** Estimation results for background input statistics using synthetic data. **a** Example membrane voltage time series with indicated spike times from a leaky I&F neuron (Eqs. (5–7)) together with membrane voltage and ISI histograms, and associated probability densities $p_V$, $p_{ISI}$ calculated using the Fokker–Planck equation (see Methods sections "Method 1: conditioned spike time likelihood" and "Method 2: derived spike rate model"). Parameter estimates are shown in (**b**). Dots denote the values of the conditioned spike time likelihoods, i.e., the factors in Eq. (1), which are points of $p_{ISI}$ here. Method 1 was used for parameter estimation. **b** Log-likelihood subtracted from the maximal value as a function of the input mean $\mu$ and standard deviation $\sigma$, based on 400 spikes from the example in (**a**), with true and estimated parameter values indicated. **c** Maximal log-likelihood across $\mu$ and $\sigma$ as a function of fixed $\tau_m$, subtracted from the maximal value across $\tau_m$ which is attained at the indicated value. **d** Mean and central 50% (i.e., 25–75th percentile) of relative errors between estimated and true parameter values for $\mu$ and $\sigma$ as a function of number of spikes $K$. Insets: empirical density of parameter estimates with true values indicated for $K = 100$ and $K = 400$. **e** top: spike rate and ISI coefficient of variation (CV) calculated analytically using the Fokker–Planck equation (lines; see Methods sections "Method 1: conditioned spike time likelihood" and "Method 2: derived spike rate model") and empirically using numerical simulations (dots) as a function of $\mu$ for different values of $\sigma$; bottom: standard deviation of estimates for $\mu$ and $\sigma$ according to the theoretical bound given by the Cramer–Rao inequality (lines; see Methods section "Calculation of the Cramer–Rao bound") and empirical values from simulations (dots) for $K = 400$. In **a**, **b**, **d**, **e** $\tau_m$ was set to the true value

cortex of awake ferrets during silence[52] (for details see Methods section "Estimating neuronal input statistics from in vivo data"). Examples of baseline data in terms of ISI histograms together with estimation results are shown in Fig. 2e, and estimated parameter values of the background input for all cells are visualized in Fig. 2f. The I&F model with fluctuating background input captures well a range of ISI histograms that characterize the baseline spiking activity of cortical neurons. Also here the I&F model appears to be the preferred one compared to a Poisson process for almost all cells according to the AIC (Fig. 2g).

**Inference of input perturbations.** We next focus on the effects of synaptic or stimulus-driven inputs at known times. Specifically, we consider $\mu(t) = \mu_0 + J\mu_1(t)$, where $\mu_0$ denotes the background mean input and $\mu_1(t)$ reflects the superposition of input pulses triggered at known times. For this scenario we apply and compare methods 1a and 2.

For an evaluation on synthetic data we model $\mu_1(t)$ by the superposition of alpha functions with time constant $\tau$, triggered at known event times with irregular inter-event intervals. We estimate the perturbation strength $J$ as well as $\tau$, which determines the temporal extent over which the perturbation acts. Estimation accuracy for a range of perturbation strengths is shown in Fig. 3a, b. Note that the input perturbations are relatively weak, producing mild deflections of the hidden membrane voltage which are difficult to recognize visually in the membrane voltage time series in the presence of noisy background input (Fig. 3a). Both methods perform comparably well for weak input perturbations. As $|J|$ increases the estimation accuracy of method 2 increases, whereas that of method 1a decreases (Fig. 3b) because it is based on a weak coupling approximation.

We further assessed the sensitivity of our estimation methods for the detection of weak input perturbations, and considered a model-free method based on cross-correlograms (CCGs) between spike trains and perturbation times (Fig. 3c, d) for comparison. Briefly, detection sensitivity was measured by the fraction of significant estimates of $J$ and CCG extrema for positive lags, respectively, compared to the estimated values and CCGs without perturbations from large numbers of realizations (for details see Methods section "Modeling input perturbations"). The model-free approach estimates the probability that the input and the spike train are coupled, but does not provide additional information on the shape of that coupling. Both model-based estimation methods are more sensitive in detecting weak perturbations than the model-free approach, with method 1a expectedly performing best (Fig. 3d).

For additional benchmarks we considered the approach from ref. [62] (Supplementary Fig. 3). We compared estimation accuracy for the parameters of perturbations (Supplementary Fig. 3a), background input and the membrane time constant (Supplementary Fig. 3b), detection sensitivity (Supplementary Fig. 3c) as well as computation time (Supplementary Fig. 3d). Both of our methods are clearly more accurate in terms of parameter estimation (Supplementary Fig. 3a, b) and thus, reconstruction (Supplementary Fig. 3e). As a result, detection sensitivity is improved (Supplementary Fig. 3c), while computation time is strongly reduced (Supplementary Fig. 3d).

Using our model we then examined how the statistics of the background input affect the ability to detect weak input perturbations. For this purpose we analytically computed the change in expected ISI caused by a weak, brief input pulse (Supplementary Fig. 4). That change depends on the time within

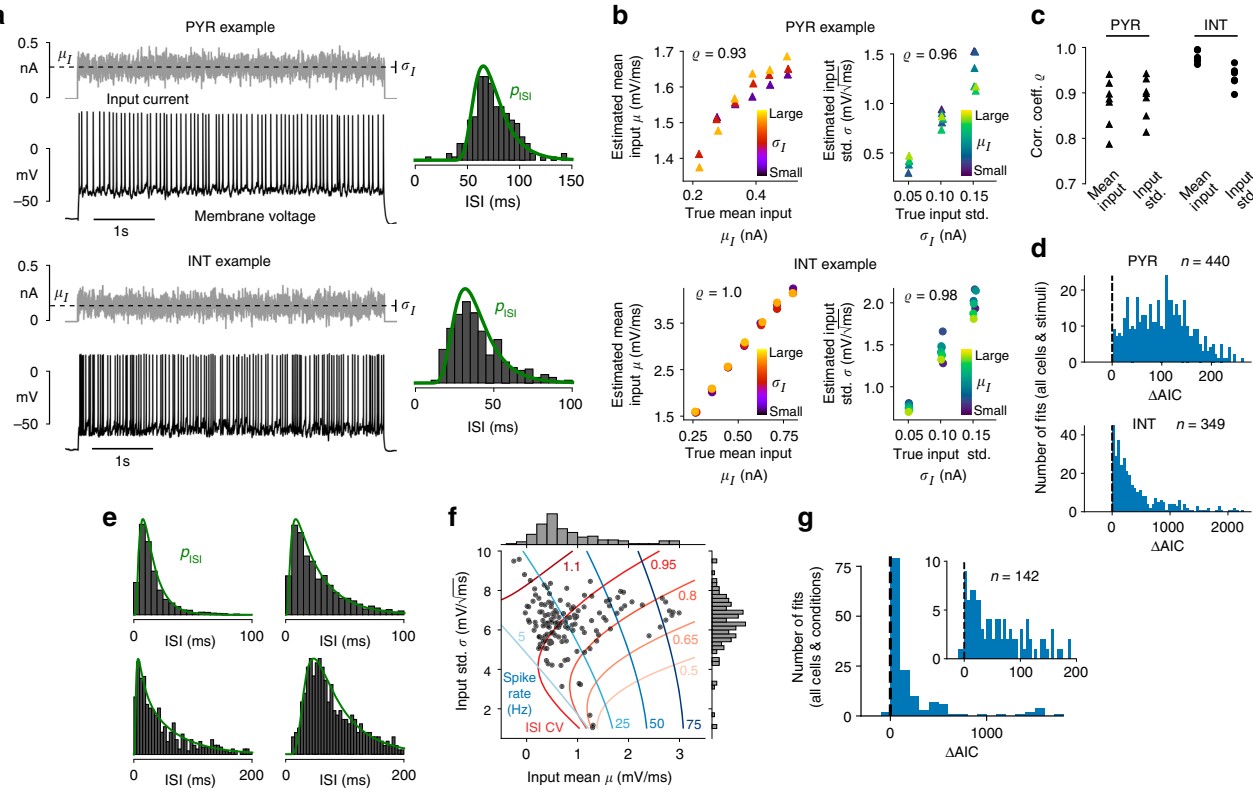

**Fig. 2** Estimation results for background input statistics using in vitro and in vivo data. **a** Examples of recorded membrane voltage in response to fluctuating input with indicated mean $\mu_I$ and standard deviation $\sigma_I$ (vertical bar marks $\mu_I \pm \sigma_I$) together with ISI histogram and density $p_{ISI}$ that corresponds to the fitted leaky I&F model for a PYR (top) and an INT (bottom). Method 1 was used for parameter estimation. **b** Estimated input parameters ($\mu, \sigma$) versus empirical input statistics ($\mu_I, \sigma_I$) with Pearson correlation coefficient $\varrho$ indicated for an example PYR and INT each. Note that since we considered only spikes, and not the membrane potential, we did not estimate the input resistance and rest potential; therefore, the input parameters were defined up to arbitrary offset and scale factors. **c** $\varrho$ for input mean and standard deviation for all seven PYRs and six INTs. **d** Histograms of AIC difference ($\Delta$AIC) between the Poisson and I&F models across all stimuli for all PYRs (top) and all INTs (bottom). **e** Examples of ISI histograms from in vivo recordings and densities $p_{ISI}$ from fitted I&F neurons. **f** Estimates of parameters for the background input together with contour lines of equal spike rate and ISI CV calculated from $p_{ISI}$. **g** Histogram of AIC difference between the Poisson and I&F models across all cells and conditions

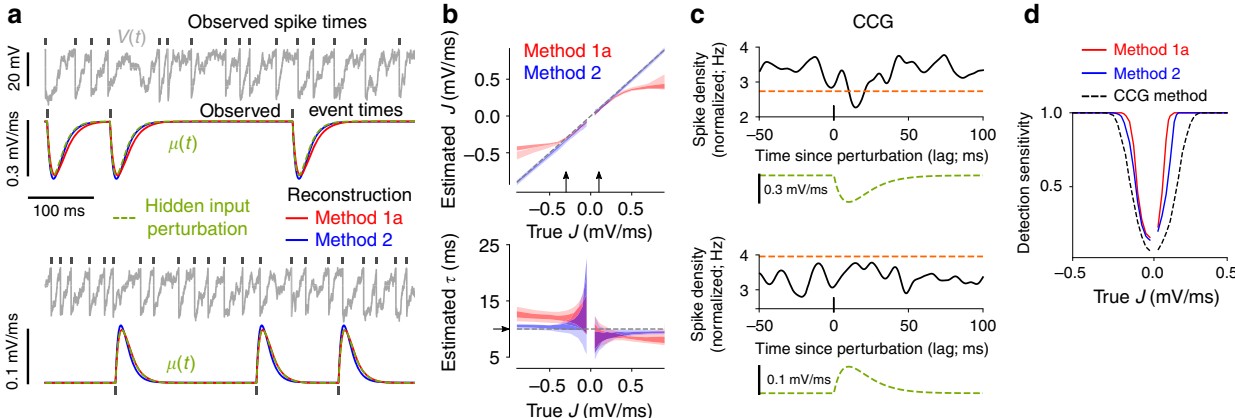

**Fig. 3** Estimation results for input perturbations using synthetic data. **a** Example membrane voltage and mean input time series (true and reconstructed ones using the estimated parameter values) for weak excitatory (lower panel) and somewhat stronger inhibitory (upper panel) perturbations. Methods 1a and 2 were used for parameter estimation. **b** Central 50% (i.e., 25–75th percentile) of estimates for $J$ and $\tau$ as a function of true $J$ for two spike train lengths (100 s, light shade; 500 s, dark shade). Arrows indicate the parameter choices in (**a**). **c** Normalized CCGs (i.e., spike density curves aligned to perturbation onset times) with significance threshold (orange dashed line) for the indicated mean input perturbations, corresponding to the examples in (**a**). The CCGs are normalized such that their integral over all lags equals one. **d** Detection sensitivity as a function of $J$ computed from the estimation methods (solid lines) or using CCGs (dashed line) based on spike trains of length 100 s (cf. **c**). For details see Methods section "Modeling input perturbations"

the ISI at which the perturbation occurs (Supplementary Fig. 4a). By comparing the maximal relative change of expected ISI across parameter values for the background input we found that a reduction of spiking variability within biologically plausible limits (ISI CV $\gtrsim 0.3$) typically increases detectability according to this measure (Supplementary Fig. 4b).

We validated our inference approach using whole-cell recordings of cortical pyramidal cells that received injections of a fluctuating current and an additional artificial excitatory postsynaptic current (aEPSC)[50]. The fluctuating current was calibrated for each cell to produce ~5 spikes/s and membrane voltage fluctuations of maximal ~10 mV amplitude. aEPSC bumps with 1 ms rise time and 10 ms decay time were triggered at simulated presynaptic spike times with rate 5 Hz, and their strength varied in different realizations (for details see Methods section "In vitro ground truth data on input perturbations").

We applied methods 1a and 2 to detect these artificial synaptic inputs based on the presynaptic and postsynaptic spike times only, and compared the detection performance with that of a CCG-based method. Specifically, input perturbations were described by delayed pulses for method 1a (which yields improved computational efficiency) and alpha functions for method 2 (cf. evaluation above). We quantified how much data is necessary to detect synaptic input of a given strength. For this purpose we computed the log-likelihood ratio between the I&F model with ($J \neq 0$) and without ($J = 0$) coupling in a cross-validated setting, similarly to the approach that has been applied to this dataset previously using a Poisson point process GLM[50]. For the model-free CCG method we compared the CCG peak for positive lags to the peaks from surrogate data generated by distorting the presynaptic spike times with a temporal jitter in a large number of realizations. Detection time is then defined respectively as the length of data at which the model with coupling provides a better fit than the one without according to the likelihood on test data[50] (Fig. 4a), and at which the CCG peak crosses the significance threshold (Fig. 4b; for details see Methods section "In vitro ground truth data on input perturbations").

All three methods were able to successfully detect most aEPSC perturbations, and method 2 required the least data for detection (Fig. 4c; overall reduction in detection time, also compared to the GLM used previously: Fig. 2F, G in ref. [50] is directly comparable to Fig. 4a, c). We further considered an alternative detection criterion for the I&F model, based on surrogate data as used for the CCG method. This approach thus enables a more direct comparison. Comparing the detection performance for fixed recording duration across all perturbation strengths and available cells shows that method 1a is more sensitive than the CCG method (success rate 0.86 vs. 0.78, Fig. 4d). We conclude that both methods, 1a with delayed pulses and 2 with an alpha function kernel, are well suited to detect weak aEPSCs in this in vitro dataset, and their sensitivity is increased compared to a model-free CCG method.

**Inference of synaptic coupling**. In the previous section we showed that we can successfully estimate the perturbations in the spiking of an individual neuron that may be elicited by inputs from another neuron. We now turn to estimating synaptic couplings in a network. We consider the situation in which we have observed spike trains of $N$ neurons. We fit these data using a network model in which each neuron $i$ receives independent fluctuating background input with neuron-specific mean $\mu_i(t)$ and variance $\sigma_i^2$, and neurons are coupled through delayed current pulses which cause postsynaptic potentials of size $J_{i,j}$ with time delay $d_{i,j}$, for $i, j \in \{1, \ldots, N\}$. The mean background input may vary over time to reflect large amplitude variations in the external

drive (and thus, spiking activity) that are slow compared to the fast input fluctuations captured by the Gaussian white noise process. Our aim is therefore to estimate the coupling strengths in addition to the statistics of background inputs caused by unobserved neurons.

We collect the observed spike times of all $N$ neurons in the set D and separate the matrix of coupling strengths $\mathbf{J}$ from all other parameters in $\boldsymbol{\theta}$ for improved clarity. Since $\mu_i(t)$ is assumed to vary slowly we approximate it by one value across each ISI. The overall mean input for neuron $i$ across the $k$th ISI can therefore be expressed as $\mu_i^k + \sum_{j=1}^{N} J_{i,j} \mu_j^1(t)$, where $J_{i,j} \mu_j^1(t)$ describes the synaptic input for neuron $i$ elicited by neuron $j$ taking the delay $d_{i,j}$ into account. The likelihood $p(\mathrm{D}|\boldsymbol{\theta}, \mathbf{J})$ can be factorized into conditioned spike time likelihoods, where each factor is determined by the parameters in $\boldsymbol{\theta}$ together with a specific subset of all coupling strengths and knowledge of the spike times that we have observed. Assuming reasonably weak coupling strengths, each of these factors can be approximated by the sum of the conditioned spike time likelihood in absence of input perturbations and a first order correction due to synaptic coupling (cf. Eq. (2)) to obtain

$$p(\mathrm{D}|\boldsymbol{\theta}, \mathbf{J}) \approx \prod_{i=1}^{N} \prod_{k=1}^{K_i-1} p_0(t_i^{k+1}|t_i^k, \boldsymbol{\theta}) + \sum_{j=1}^{N} J_{i,j}\, p_1(t_i^{k+1}|t_i^k, \mu_j^1[t_i^k, t_i^{k+1}], \boldsymbol{\theta}),$$

$$(4)$$

where $t_i^k$ denotes the $k$th of $K_i$ observed spike times and $\boldsymbol{\theta}$ contains all parameters, including $\mu_i^k$ (except for $\mathbf{J}$). Note that the mean input perturbations $\mu_j^1$ depend on the spike times of neuron $j$ taking the delay $d_{i,j}$ into account. The approximation (4) implies that an individual synaptic connection on one neuron has a negligible effect for the estimation of a connection on another neuron, which is justified by the assumption of weak coupling. This allows for the application of method 1a, by which the likelihood can be calculated in an efficient way (for details see Methods section "Network model and inference details").

We first evaluated our method on synthetic data for small ($N = 10$) as well as larger ($N = 50$) fully observed networks of neurons with constant background input statistics. The number of parameters inferred per network, which include mean and standard deviation of background inputs, coupling strengths and delays, excluding self-couplings, amounts to $2N^2$. The estimated parameter values show a remarkable degree of accuracy (Fig. 5a, b left and Supplementary Fig. 5a).

In a more realistic scenario, the $N$ recorded neurons belong to a larger network that is subsampled through the measurement process. The unobserved neurons therefore contribute additional, hidden inputs. In the fitted model, the effect of these unobserved neurons on neuron $i$ is absorbed in the estimated parameters ($\mu_i$, $\sigma_i$) of the background noise. Specifically, the total external input to neuron $i$, originating from a large number $M_i$ of unobserved neurons whose spike trains are represented by independent Poisson processes with rates $r_j$, can be approximated for reasonably small coupling strengths with a background noise of mean $\mu_i = \sum_{j=N+1}^{N+M_i} J_{i,j} r_j$ and variance $\sigma_i^2 = \sum_{j=N+1}^{N+M_i} J_{i,j}^2 r_j$. This is the classical diffusion approximation[63]. Because of shared connections from unobserved neurons, the inputs received by the different observed neurons are in general correlated, with correlation strengths that depend on the degree of overlap between the unobserved presynaptic populations. Although the model we fit to the observed data assumes uncorrelated background inputs, the effects of input correlations on the neuronal ISI distributions are approximated in the estimation of $\mu_i$ and $\sigma_i$.

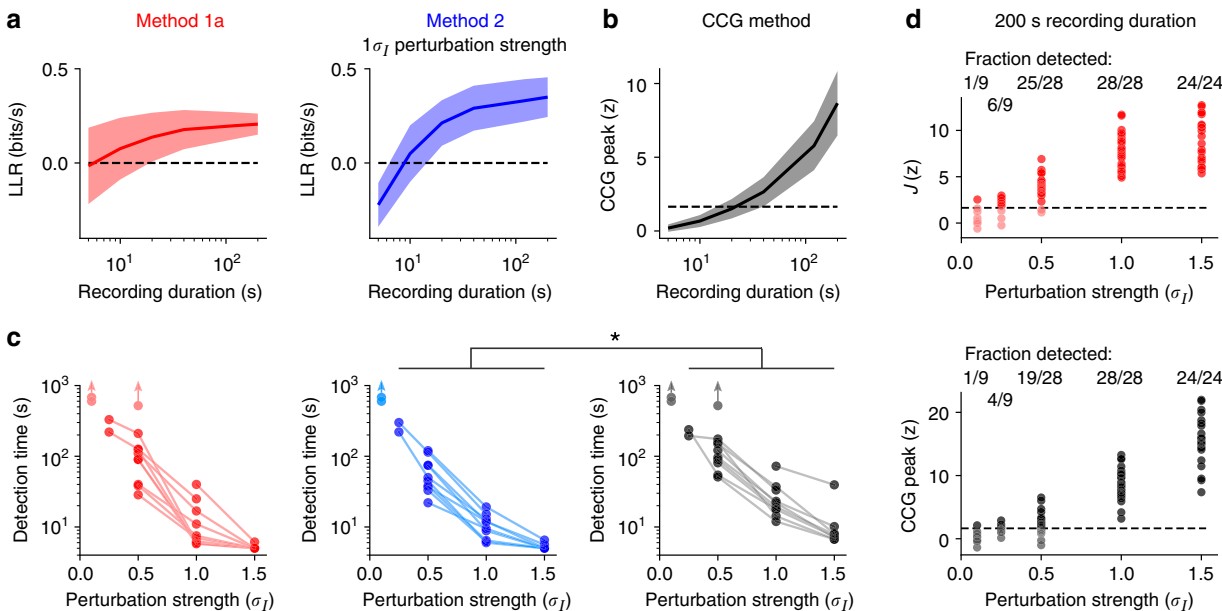

**Fig. 4** Estimation results for artificial synaptic input perturbations using in vitro data. **a** Log-likelihood ratio (LLR) of the I&F model with/without coupling on test data (10-fold cross-validation) as a function of recording duration for fixed aEPSC strength (in unit of $\sigma_I$, the standard deviation of the fluctuating input current) using methods 1a (left) and 2 (right). Shaded areas in **a** and **b** denote mean ± standard deviation across all cells. LLR > 0 indicates successful detection. **b** CCG peak $z$-score as a function of recording duration for aEPSC strength 1 $\sigma_I$. Dashed line indicates the significance threshold (95th percentile of the standard normal distribution). **c** Detection time for each cell as a function of aEPSC strength (in unit of $\sigma_I$) for method 1a (left), 2 (center), and the CCG method (right). Lines connect detection results for the same cell, light dots denote misses. In those cases, the detection time, if it exists, must be longer than the recording length, as indicated by arrows. * significant reduction of detection time ($p = 0.0276$, two-tailed paired $t$ test, sample size $n = 32$). **d** Estimated perturbation strength $z$-score using method 1a (top) and CCG peak $z$-score (bottom), both as a function of aEPSC strength for 200 s recording length. Recordings were split into 200 s segments with up to 40 s overlap. Dashed lines indicate the significance threshold (cf. **b**). Dark dots mark successful detections

To assess the influence of correlations due to unobserved common inputs, we first fitted our model to data generated from a network with correlated background inputs. The estimation accuracy of synaptic strengths is still good in case of weak correlations of the external input fluctuations (correlation coefficient $c = 0.1$ for each pair of observed neurons; Fig. 5b right). Empirical values for such noise correlations, measured in experimental studies by the correlation coefficient of pairwise spike counts on a short timescale, are typically very small[36,64,65].

We next tested our method on data generated by explicitly subsampling networks of randomly connected neurons, and compared the results with those from two classical approaches: a model-free CCG method (cf. Results section "Inference of input perturbations") and a method based on a Poisson point process GLM that is well constrained for the synthetic data[66] (for details see Methods section "Network model and inference details"). We chose a GLM that is tailored to capture the spiking dynamics of the (ground truth) I&F network model with a minimal number of parameters.

First, we considered classical networks of 800 excitatory and 200 inhibitory neurons[33] with connection probabilities of 0.1 for excitatory connections, 0.4 for inhibitory connections, and a plausible range of coupling strengths. Inference results from partial observations ($N = 20$) show that the I&F method outperforms the other two approaches on accuracy of both estimated coupling strengths and detection of connections (Fig. 5c–g, for different recording lengths see Supplementary Fig. 5b).

To gain more insight into the challenges caused by shared input from unobserved neurons and how they affect the different methods, we then varied connection probability and delay. We considered networks with equal numbers of excitatory and inhibitory neurons to ensure equal conditions for the estimation

of the two connection types (Fig. 5h–k and Supplementary Fig. 5c–f). For relatively sparse, subsampled networks (connection probability 0.1, $N = 20$ observed neurons out of 1000) all three methods perform well, and the I&F method shows only a slight improvement in terms of correlation between true and estimated coupling strengths and detection accuracy (Fig. 5h, for detailed inference results see Supplementary Fig. 5c). The inference task becomes increasingly difficult as connectivity increases (connection probability 0.3, see Fig. 5i and Supplementary Fig. 5d). In this case the correlations between spike trains of uncoupled pairs of neurons are clearly stronger on average, particularly at small time lags, which renders CCGs unsuitable to distinguish the effects of numerous synaptic connections from uncoupled pairs. As a consequence, the number of false positives and misclassified inhibitory connections increases. Hence, the accuracy of the CCG method is substantially reduced. This also causes strongly impaired accuracy of the GLM method, especially with respect to detection of synapses, whereas our I&F method appears to be much less affected. Increased synaptic delays in such networks lead to improved inference results for all three methods (Fig. 5j and Supplementary Fig. 5e). Intuitively, this shifts the effects of connections in the CCG to larger lag values and thus away from the peak at zero lag caused by correlated inputs (Fig. 5j left). Nevertheless, also in this case the I&F method remains the most accurate one. We further considered small spike train distortions (using a temporal jitter) to mimic strong measurement noise, which naturally caused an overall reduction in accuracy but most strongly affected the CCG method (Fig. 5k and Supplementary Fig. 5f).

Finally, we would like to note that the connectivity is not the only determinant of noise correlations in random networks; an increase of coupling strengths also caused increased spike train

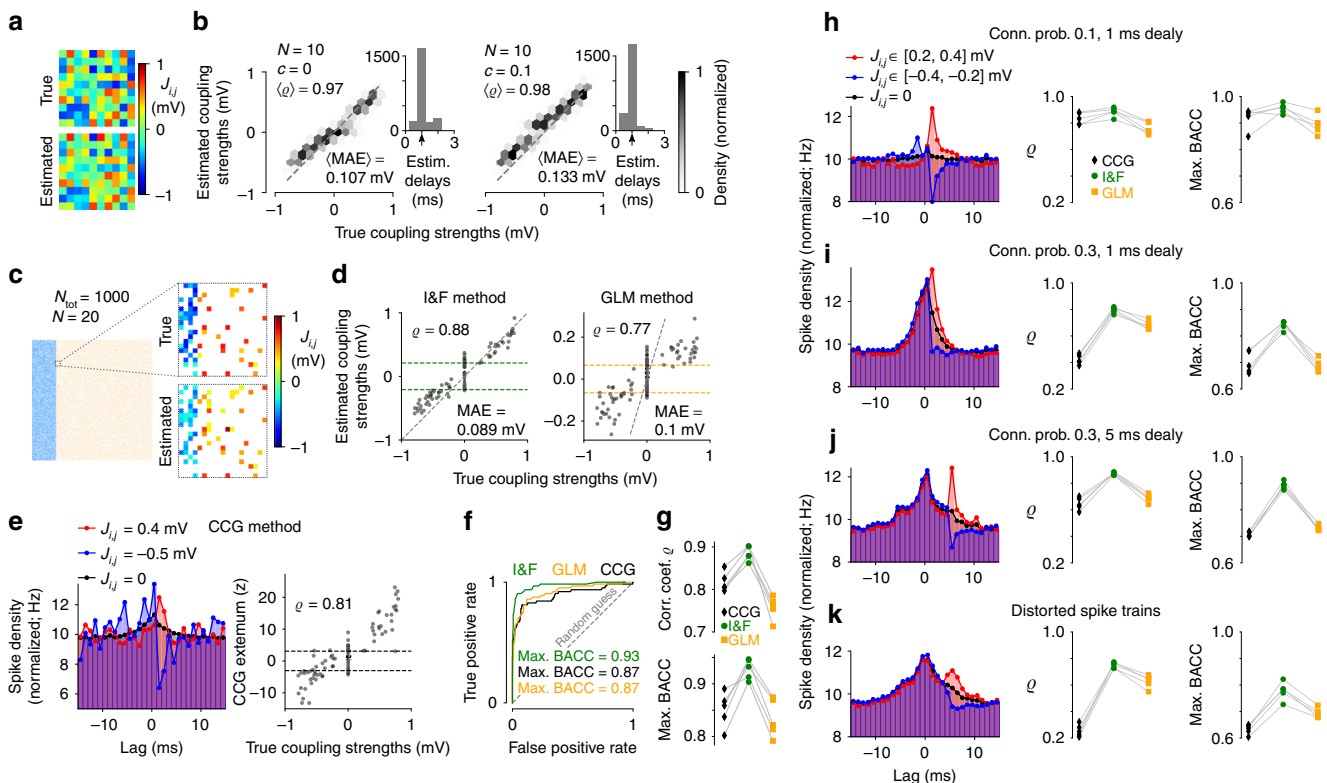

**Fig. 5** Estimation results for synaptic coupling strengths using synthetic data. **a** Example coupling matrix using 5 min long spike trains from $N = 10$ neurons and estimation method 1a. **b** Density of estimated vs. true coupling strengths from 25 networks of $N = 10$ neurons for uncorrelated ($c = 0$, left, as in **a**) and correlated ($c = 0.1$, right) external input fluctuations. Average Pearson correlation coefficient $\langle \varrho \rangle$ and mean absolute error $\langle \text{MAE} \rangle$ between true and estimated coupling strengths are indicated. Insets: histogram of estimated delays across all networks, arrow marks the true (global) delay. **c** True and estimated coupling strengths of $N = 20$ neurons from a randomly coupled network of $N_{tot} = 1000$ (800 excitatory, 200 inhibitory) neurons with connection probabilities of 0.1 for excitatory and 0.4 for inhibitory synapses, heterogeneous synaptic strengths and statistics of external inputs; $c = 0$, delay 1 ms. **d–f** Inference results from the I&F method compared to GLM- and CCG-based methods for the network in (**c**). **d** Estimated vs. true coupling strengths for the I&F method (left, as used in **c**) and the GLM method (right). Dashed gray lines mark identity. A discrimination threshold for the presence of connections was applied (indicated by green and orange dashed lines; for details see Methods section "Network model and inference details"). **e** Left: spike train CCGs for an excitatory connection (red), an inhibitory connection (blue) and averaged over all uncoupled pairs (black); right: CCG extremum $z$-score for positive lags vs. true coupling strengths, with discrimination threshold indicated. **f** Receiver operating characteristic (ROC) curves for the detection of synapses, i.e., true-positive rate vs. false-positive rate as the discrimination threshold varies, for the three methods. Indicated maximal balanced accuracy (BACC) values correspond to the thresholds visualized in (**d**) and (**e**), right. Dashed line represents random guessing. **g** Pearson correlation coefficient $\varrho$ (top, cf. **d** and **e**, right) and max. BACC (cf. **f**) for the three methods using five different networks as in (**c**). Results from the same network are connected by gray lines. **h–k** Effects of connection probability and delay for subsampled networks of 500 excitatory and 500 inhibitory neurons. Left column: spike train CCGs averaged over subsets of excitatory (red) and inhibitory (blue) connections as indicated, and over all uncoupled pairs (black). Center and right columns: $\varrho$ and max. BACC for the three methods (as in **g**). Connection probability and delay values are indicated in (**h–j**); **k** setting as in (**j**), but spike trains were perturbed by a temporal jitter (random values in $[-1, 1]$ ms). Other parameter values as in (**c**). Results in (**h–k**) are from five networks each. In (**c–k**), 10 min long recordings were used

correlations for uncoupled pairs (Supplementary Fig. 6). For rather sparse networks (connection probability 0.1) the benefit of stronger couplings outweighs that disadvantage for inference (Supplementary Fig. 6a). However, for slightly increased connectivity (connection probability 0.2) the noise correlations are strongly amplified, leading to an increase in mainly false positives and misclassified inhibitory connections (Supplementary Fig. 6b, in comparison to Fig. 5i and Supplementary Fig. 5d).

In sum, our approach yields accurate inference results for subsampled networks as long as the correlations between the hidden inputs, due to shared connections from unobserved neurons, are not too large. In particular, it outperforms classical CCG-based and GLM-based methods.

We validated our inference of synaptic coupling using simultaneous extracellular recordings and juxtacellular stimulations of hippocampal neuronal ensembles in awake mice[51].

Following the approach developed in ref. [51], we estimated connectivity by applying the I&F method to spontaneous, extracellularly recorded spiking activity, and assessed the accuracy of our estimates by comparison with ground truth data. Ground truth connectivity was obtained by evoking spikes in single PYRs juxtacellularly using short current pulses, while recording extracellular spike trains of local INTs (for an example see Fig. 6a). Ground truth values for the presence and absence of synaptic connections were derived from spike train CCGs using the evoked presynaptic spikes, taking into account co-modulation caused by common network drive (for details see Methods section "In vivo ground truth data on synaptic connections").

An important aspect of these data is that spontaneous activity appeared to be highly nonstationary, so that the spike trains of the recorded neurons were typically co-modulated. To infer synaptic couplings from the spontaneous spike trains with our

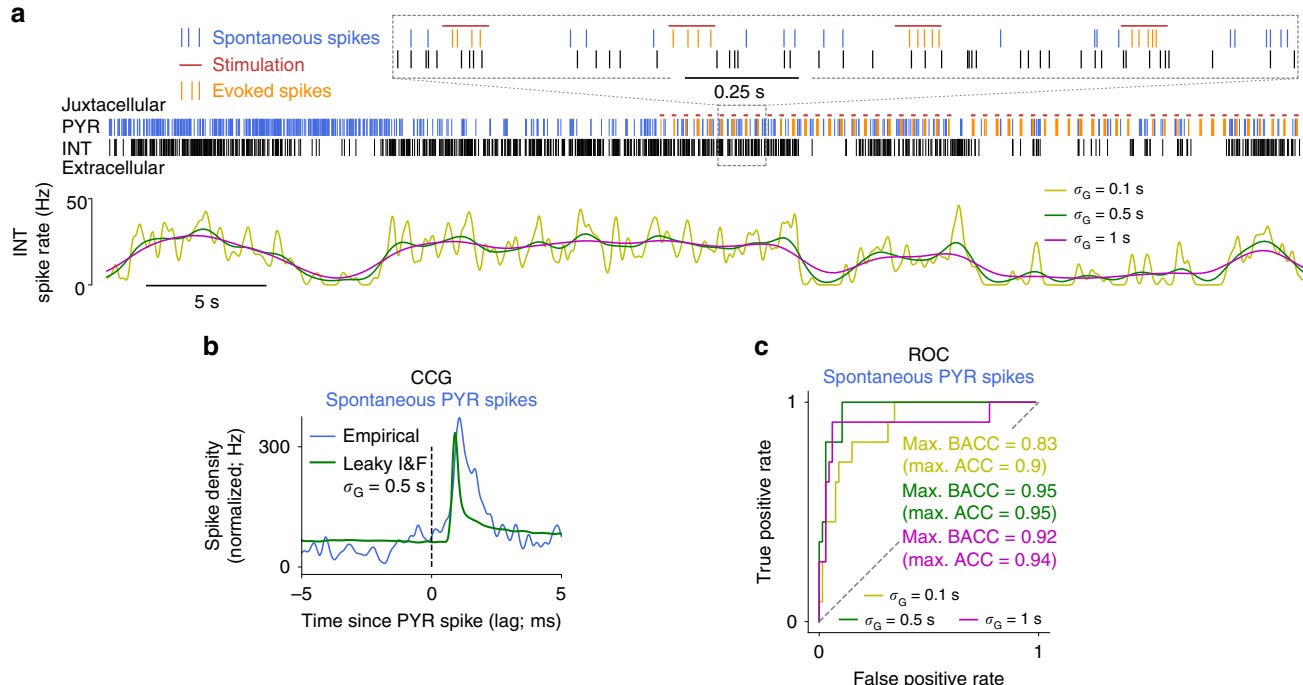

**Fig. 6** Estimation results for synaptic couplings using in vivo data. **a** Example spike trains of a juxtacellularly recorded PYR and an extracellularly recorded INT, with stimulation pulses indicated (top panel). Instantaneous spike rate of the INT computed via kernel density estimation using a Gaussian kernel with width $\sigma_G$ as indicated (bottom panel). **b** CCG between spontaneous PYR and INT spike times and as computed from an I&F neuron fitted to the INT spike train with estimated coupling strength and delay, corresponding to the example in (**a**). Note that synaptic coupling was modeled using delayed, transient pulses. Method 1a was used for parameter estimation. **c** ROC curves across 78 PYR-INT pairs using the spontaneous PYR spikes (up to 1000 per pair) for three different values of $\sigma_G$. Maximal (balanced) accuracy values are indicated

model-based approach, we accounted for network co-modulation in two ways: (1) through small temporal perturbations of the PYR spike times, used to compute coupling strength $z$-scores; (2) through estimated variations of the background mean input for the (potentially postsynaptic) INTs, that is, $\mu_i^k$ in Eq. (4) varied between ISIs. These variations were inferred from the instantaneous spike rate, which typically varied at multiple timescales over the duration of the recordings that lasted up to ~2 h (Fig. 6a). We, therefore, estimated the variations of mean input at three different timescales separately and inferred synaptic couplings for each of these (see Methods section "In vivo ground truth data on synaptic connections").

Although spontaneous activity was highly nonstationary, our inference of the connectivity appeared to be very accurate. Comparisons with ground truth estimates demonstrated accuracy of up to 0.95 (for the intermediate timescale variant; Fig. 6c). Moreover, reducing the number of spikes used for inference did not lead to an appreciable decrease of reproduction accuracy (Supplementary Fig. 7a). Nevertheless, using instead a model-free CCG method on the spontaneous spike trains yielded comparable detection accuracy (Supplementary Fig. 7b, for an example CCG see Fig. 6b). This fact and the observation that the timescale affects detection performance only weakly (cf. Fig. 6c) may be explained by the large signal-to-noise ratio in this dataset (Supplementary Fig. 7c), as the focus in ref. [51] was on strong connections. We would also like to remark that a GLM-based method was previously applied and compared to the CCG-based method on these data[51], resulting in similar detection accuracy at strongly increased computational demands (200–400 s GPU computing time for a GLM fit, depending on the recording length, vs. 20 ms CPU computing time for the CCG analysis).

**Inference of neuronal adaptation**. We next extend the model neurons to account for spike rate adaptation—a property of many types of neurons, including pyramidal cells[67–69]. It can be observed by a gradual change in spiking activity following an immediate response upon an abrupt change of input strength, as shown in Fig. 7a, d. This behavior is typically mediated by a calcium-activated, slowly decaying transmembrane potassium current, which rapidly accumulates when the neuron spikes repeatedly[69,70]. In the extended I&F neuron model[14,55] this adaptation current is represented by an additional variable $w$ that is incremented at spike times by a value $\Delta w$, exponentially decays with slow time constant $\tau_w$ in between spikes, and subtracts from the mean input, acting as a negative feedback on the membrane voltage (Fig. 7a, see Methods section "Modeling spike rate adaptation").

In contrast to classical I&F neurons, in the generalized model with adaptation, spiking is not a renewal process: given a spike time $t_k$ the probability of the next spike depends on all previous spike times. That dependence is however indirect, as it is mediated through the effective mean input $\mu(t) - w(t)$ across the ISI $[t_k, t_{k+1}]$. This effective mean input can be explicitly expressed using the parameter values in $\boldsymbol{\theta}$ together with the observed spike times, and then inserted in Eq. (1) for estimation. Here, method 1 is best suited and can be applied efficiently by exploiting the fact that $w$ varies within ISIs in a rather stereotyped way (for details see Methods section "Modeling spike rate adaptation").

We first evaluated the inference method using simulated ground truth data for constant statistics of the background inputs (cf. Results section "Inference of background inputs"). An example of the membrane voltage and adaptation current time series is depicted in Fig. 7b. The true values for the adaptation parameters (the strength $\Delta w$ and time constant $\tau_w$) are well

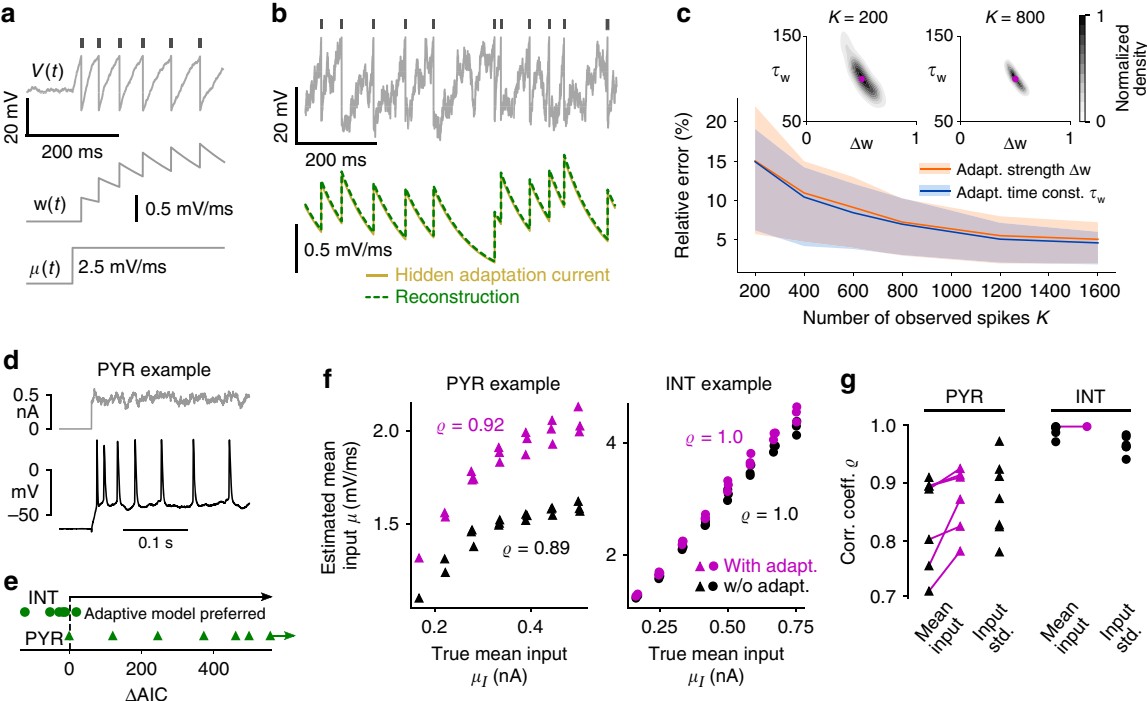

**Fig. 7** Estimation results for neuronal adaptation using synthetic and in vitro data. **a** Membrane voltage with indicated spike times (top) and adaptation variable (center) of an adaptive leaky I&F neuron in response to a step of mean input (bottom, for small input noise intensity $\sigma$). **b** Example membrane voltage and true adaptation current time series as well as the reconstruction using estimated adaptation parameters $\Delta w$, $\tau_w$ (based on 1000 spikes) and the observed spike times. The adaptation current is considered as normalized by the (unknown) membrane capacitance and therefore in units of mV/ms. Method 1 was used for estimation. **c** Mean and central 50% (i.e., 25–75th percentile) of relative errors between estimated and true values for $\Delta w$ and $\tau_w$ as a function of number of spikes $K$. Insets: empirical density of estimated parameter values with true values indicated for $K = 200$ and $K = 800$. **d** Example recorded membrane voltage in response to an injected noisy step current showing spike rate adaptation. **e** AIC difference ($\Delta$AIC) between the nonadaptive and adaptive leaky I&F models for all seven PYRs and six INTs. **f** Estimated mean input $\mu$ as a function of empirical mean input $\mu_I$ for the adaptive and nonadaptive models (magenta and black symbols, respectively) for two example cells. $\varrho$ denotes Pearson correlation coefficient. **g** $\varrho$ for input mean and standard deviation for the two models and all neurons. Estimation results in (**e**–**g**) were obtained using 15 s long stimuli (corresponding to three repetitions of 5 s stimuli with equal statistics)

recovered and we obtain an accurate estimate of the adaptation current as it evolves over time. Here, too, the estimation accuracy depends on the number of observed spikes (Fig. 7c) and relative errors are on average less than 10% for 500 spikes.

Comparisons of our method with the approach from ref. [62] on simultaneous inference of adaptation and background input parameters show clear improvements in terms of estimation accuracy (Supplementary Fig. 8a, b, d) and computation time (Supplementary Fig. 8c).

To validate our inference method for adaptation parameters, we used the recordings of neurons stimulated by noise currents that we examined in Results section "Inference of background inputs". Several cells, predominantly PYRs, exhibited clear spike rate adaptation (for an example see Fig. 7d). Accordingly, the adaptive I&F model yielded a clearly improved fit compared to the nonadaptive model for all but one PYRs as shown by the AIC (Fig. 7e; for details see Methods section "In vitro ground truth data on neuronal input statistics"). On the other hand, for all except one INTs the nonadaptive model turned out to be the preferred one, which is consistent with the observation that INTs generally exhibit little spike rate adaptation compared to PYRs[71].

We examined the mean input estimates from the adaptive model in comparison to the nonadaptive model for the neurons where the adaptive model was preferred (Fig. 7e, f). For all of those cells, including adaptation increased the correlation coefficient between estimated and empirical mean input. The remaining room for improvement of this correlation for PYRs

indicates that there are likely multiple adaptation mechanisms (with different timescales) at work[71]. Note that the intrinsic adaptation current effectively subtracts from the mean input (but does not affect the input standard deviation). Indeed, the presence of adaptation in the model insignificantly affects the correlation coefficient between estimated and empirical input standard deviation (Supplementary Fig. 9). This can be explained by the fact that the adaptation variable varies slowly compared to the membrane voltage and typical mean ISI (estimated $\tau_w > 4\tau_m$ on average); therefore, it affects the ISI mean more strongly than its variance which is predominantly influenced by the fast input fluctuations (parameter $\sigma$).

## Discussion

We presented efficient, statistically principled methods to fit I&F circuit models to single-trial spike trains, and we evaluated and validated them extensively using synthetic, in vitro and in vivo ground truth data. Our approach allows to accurately infer hidden neuronal input statistics and adaptation currents as well as coupling strengths for I&F networks. We demonstrated that (1) the mean and variance of neuronal inputs are well recovered even for relatively short spike trains; (2) for a sufficient, experimentally plausible number of spikes, weak input perturbations triggered at known times are detected with high sensitivity, (3) coupling strengths are faithfully estimated even for subsampled networks, and (4) neuronal adaptation strength and timescale are accurately

inferred. By applying our methods to suitable electrophysiological datasets, we could successfully infer the statistics of in vivo-like fluctuating inputs, detect input perturbations masked by background noise, reveal intrinsic adaptation mechanisms, and reconstruct in vivo synaptic connectivity.

Previously several likelihood-based methods related to ours have been proposed, considering uncoupled I&F neurons with[61,62,72] or without adaptation[73,74] for constant[73] or time-varying[61,62,72,74] input statistics. All of these methods employ the Fokker–Planck equation and numerically calculate the spike train likelihood with high precision[73] or using approximations[61,62,72,74]. We benchmarked our methods against that from[61,62], which was applicable to the estimation of single neuron parameters and for which an efficient implementation is available (cf. Supplementary Methods Section 3). Our methods clearly outperformed the previous one in terms of estimation accuracy as well as computation time, owing to optimized numerical discretization/interpolation schemes (method 1) and effective approximations (method 2). Notably, our methods extend these previous approaches, centered on inference for single leaky I&F models, to the estimation of synaptic coupling in networks of generalized I&F neurons.

In the absence of likelihoods, methods for parameter fitting typically involve numerical simulations and distance measures that are defined on possibly multiple features of interest[46–48,75,76]. Evolutionary algorithms[46,47,75], brute-force search[48] or more principled Bayesian techniques[76] are then used to minimize the distances between observed and model-derived features. While these likelihood-free, simulation-based methods can be applied to more complex models they exhibit disadvantages: the distance measures usually depend on additional parameters and their evaluation depends on the particular realization of noise or randomness considered in the model. Optimization can therefore be exceedingly time-consuming. Furthermore, the principled, likelihood-based tools for model comparison (such as AIC and log-likelihood ratio) are not applicable in this case.

Here, we directly estimated synaptic coupling strengths for leaky I&F networks with fluctuating external inputs from observed spike trains. Our method is conceptually similar to those presented in[77,78], but does not rely on an approximation of the spike train likelihood that assumes vanishing[77] or small[78] amplitudes of input fluctuations. Strongly fluctuating inputs are typically required to produce in vivo-like spiking statistics.

Our approach outperformed a straightforward, model-free method based on CCGs as well as an approach based on a phenomenological, point process GLM[66]. This does not imply that GLM-based methods are generally less accurate in inferring synaptic connectivity: point process GLMs are flexible models which can be designed and optimized to fit the observed spike trains well[2,3,79,80]. However, that approach is prone to overfitting unless strong constraints or regularization are enforced[3,5,66,80,81]. An advantage of our approach in this respect is that the basic mechanistic principles included in I&F models provide a natural regularization and reduce the number of model parameters, which strongly reduces the risk of overfitting. Note that our method 2 is essentially based on mapping I&F models to simplified, constrained GLM-like models[57,58,82].

Alternative approaches to infer connectivity from spike trains, other than those addressed above, have employed models of sparsely and linearly interacting point processes[83], or have been designed in a model-free manner[51,84,85], for example, using CCGs[51,84] similarly to our comparisons. A general challenge in subsampled networks arises from pairwise spike train correlations at small time lags generated by shared connections from unobserved neurons, regardless of whether a direct connection is present. These spurious correlations impair our ability to distinguish the effects of synaptic connections from those caused by correlated inputs, especially when coupling delays are small. One of the benefits of our approach is that it includes an explicit, principled mechanism to account for the effects of unobserved neurons, which are absorbed in the estimated statistics of the fluctuating background inputs. Correlated fast input fluctuations are not directly modeled, their effects are compensated for in the estimation of the background input parameters, whereas shared input dynamics on a longer timescale are explicitly captured by slow variations of the mean input for each neuron. This facilitates the isolation of pairwise synaptic interactions from common drive.

Several related studies have focused on a theoretical link between network structure and correlated spiking activity recorded from a large number of neurons, without attempting to explicitly estimate synaptic connections[86–93]. Of major relevance in this regard is the extent to which effective interactions among observed neurons are reshaped by coupling to unobserved neurons[79,94]. Current methods to estimate coupling strengths from observed spike trains may be further advanced using these theoretical insights.

Throughout this work we assumed that the mean input trajectory across an ISI can be determined using available knowledge (that is, the model parameters and observed spike times). In Results section "Inference of synaptic coupling" we extracted the variations of the mean input from estimates of the instantaneous neuronal spike rate at different timescales (cf. Fig. 6). A useful extension may be to consider a separate stochastic process that governs the evolution of the mean input, allowing to extract the most appropriate timescale from the data[95], which in turn could benefit the estimation of synaptic couplings using our approach.

I&F neurons are a popular tool for interpreting spiking activity in terms of simple circuit models (see, e.g., refs. [25–31]). Such approaches typically start by hypothesizing a structure for the underlying circuit based on available physiological information, and then examine the behavior of the resulting model as a function of the critical biophysical parameters. The model is then validated by qualitatively comparing the model output with experimental data. Specifically, the model activity is required to resemble key features of the experimental data in an extended region of the parameter space. If that is not the case, the model is rejected and a different one is sought.

An important benefit of this approach is that it provides a mechanistic interpretation and understanding of recorded activity in terms of biological parameters in a neural circuit. A major limitation is, however, that it typically relies on a qualitative comparison with the data to select or reject models. The methods presented here open the door to a more quantitative, data-driven approach, in which this class of spiking circuit models can be evaluated and compared based on their fitting performance (cf. Figs. 2d, g, 4a, c, and 7e for such comparisons) as is routinely the case for more abstract statistical models (see, e.g., ref. [4]).

## Methods

**I&F neuron models.** We consider typical I&F models subject to fluctuating inputs. The dynamics of the membrane voltage $V$ are governed by

$$\frac{dV}{dt} = f(V) + \mu(t) + \sigma \xi(t) \tag{5}$$

$$\text{if } V(t) \geq V_s \text{ then } V(t) \leftarrow V_r, \tag{6}$$

where $\mu$ is the mean input, $\sigma$ the standard deviation of the input, $\xi$ a (unit) Gaussian white noise process, i.e., $\langle \xi(t)\xi(t+\tau)\rangle = \delta(\tau)$ with expectation $\langle \cdot \rangle$, $V_s$ is the threshold (or spike) voltage and $V_r$ the reset voltage. For the leaky I&F model the function $f$ is given by

$$f(V) := -\frac{V}{\tau_m}, \tag{7}$$

where $\tau_m$ denotes the membrane time constant. It should be noted that for the methods used in this paper $f$ can be any arbitrary real-valued function. For example, in the exponential I&F model, used for Supplementary Fig. 1e, $f$ is a nonlinear function that includes an exponential term (for details see Supplementary Methods section 1). The parameter values are $V_s = 30$ mV, $V_r = 0$ mV, $\tau_m = 20$ ms, $\mu = 1.75$ mV/ms, $\sigma = 2.5$ mV/$\sqrt{\text{ms}}$ if not stated otherwise in figures or captions.

It is not meaningful to estimate all model parameters: a change of $V_s$ or $V_r$ in the leaky I&F model can be completely compensated in terms of spiking dynamics by appropriate changes of $\mu(t)$ and $\sigma$. This can be seen using the change of variables $\tilde{V} := (V - V_r)/(V_s - V_r)$. Consequently, we may restrict the estimation to $\mu(t)$, $\sigma$, $\tau_m$ and set the remaining parameters to reasonable values.

**Method 1: conditioned spike time likelihood.** It is useful to express the factors in Eq. (1), the conditioned spike time likelihoods, in terms of the ISI probability density $p_{\text{ISI}}$,

$$p(t_{k+1}|t_k, \mu[t_k, t_{k+1}], \boldsymbol{\theta}) = p_{\text{ISI}}(s_k|\mu_{\text{ISI}}[0, s_k], \boldsymbol{\theta}), \qquad (8)$$

where $s_k := t_{k+1} - t_k$ is the length of the $k$th ISI and $\mu_{\text{ISI}}$ is the mean input across that ISI given by $\mu_{\text{ISI}}[0, s_k] = \mu[t_k, t_{k+1}]$. The technical advantage of this change of variables becomes most obvious for constant $\mu$. In this case the density function $p_{\text{ISI}}$ needs to be computed only once in order to evaluate the spike train likelihood, Eq. (1), for a given parametrization $\mu$, $\boldsymbol{\theta}$.

Given a spike at $t = t_0$ the probability density of the next spike time is equal to the ISI probability density $p_{\text{ISI}}(s)$ where $s := t - t_0 \geq 0$ denotes the time since the last spike. This quantity can be approximated by numerical simulation in an intuitive way: starting with initial condition $V(t_0) = V_r$ one follows the neuronal dynamics given by Eq. (5) in each of $n$ realizations of the noise process until the membrane voltage crosses the value $V_s$ and records that spike time $t_i$ in the $i$th realization. The set of times $\{t_i\}$ can then be used to compute $p_{\text{ISI}}$, where the approximation error decreases as $n$ increases. We can calculate $p_{\text{ISI}}$ analytically in the limit $n \to \infty$ by solving the Fokker–Planck partial differential equation (PDE)[96,97] that governs the dynamics of the membrane voltage probability density $p_V(V, s)$,

$$\frac{\partial p_V}{\partial s} + \frac{\partial q_V}{\partial V} = 0, \qquad (9)$$

$$q_V := [f(V) + \mu_{\text{ISI}}(s)]p_V - \frac{\sigma^2}{2}\frac{\partial p_V}{\partial V} \qquad (10)$$

with mean input $\mu_{\text{ISI}}(s) = \mu(t)$, subject to the initial and boundary conditions

$$p_V(V, 0) = \delta(V - V_r), \qquad (11)$$

$$p_V(V_s, s) = 0, \qquad (12)$$

$$\lim_{V \to -\infty} q_V(V, s) = 0. \qquad (13)$$

The ISI probability density is then given by the probability flux at $V_s$,

$$p_{\text{ISI}}(s|\mu_{\text{ISI}}[0, s], \boldsymbol{\theta}) = q_V(V_s, s). \qquad (14)$$

In the field of probability theory $p_{\text{ISI}}$ is also known as first passage time density.

In method 1a we consider the first order approximation (2) for weak perturbations of the mean input, $\mu(t) = \mu_0^k + J\mu_1(t)$ with small $|J|$ during the $k$th ISI. In this case the $k$th spike time likelihood is expressed as

$$p_{\text{ISI}}(s_k|\mu_{\text{ISI}}[0, s_k], \boldsymbol{\theta}) \approx p_{\text{ISI}}^0(s_k|\boldsymbol{\theta}) + J\, p_{\text{ISI}}^1(s_k|\mu_{\text{ISI}}^1[0, s_k], \boldsymbol{\theta}) \qquad (15)$$

$$\mu_{\text{ISI}}[0, s_k] = \mu_{\text{ISI}}^0 + J\, \mu_{\text{ISI}}^1[0, s_k], \qquad (16)$$

where $\mu_{\text{ISI}}^1[0, s_k] = \mu_1[t_k, t_{k+1}]$ and $\boldsymbol{\theta}$ contains parameters that remain constant within ISIs, including $\mu_{\text{ISI}}^0 = \mu_0^k$.

Numerical solution schemes to compute $p_{\text{ISI}}$ (method 1) or $p_{\text{ISI}}^0$ and $p_{\text{ISI}}^1$ (method 1a) in accurate and efficient ways are provided in Supplementary Methods section 2. It should be noted that these functions do not need to be computed for each observed ISI separately; instead, we pre-calculate them for a reasonable set of trajectories $\mu_{\text{ISI}}[0, s_{\max}]$, where $s_{\max}$ is the largest observed ISI, and use interpolation for each evaluation of a spike time likelihood.

**Method 2: derived spike rate model.** Method 2 requires the (instantaneous) spike rate $r(t)$ of the model neuron described by Eqs. (5) and (6), which can be calculated by solving a Fokker–Planck system similar to Eqs. (9–13),

$$\frac{\partial p_V}{\partial t} + \frac{\partial q_V}{\partial V} = 0, \quad q_V := [f(V) + \mu(t)]p_V - \frac{\sigma^2}{2}\frac{\partial p_V}{\partial V}, \quad r(t) = q_V(V_s, t), \quad (17)$$

subject to the conditions

$$p_V(V_s, t) = 0, \quad \lim_{V \to -\infty} q_V(V, t) = 0, \qquad (18)$$

$$\lim_{V \searrow V_r} q_V(V, t) - \lim_{V \nearrow V_r} q_V(V, t) = q_V(V_s, t), \qquad (19)$$

where Eq. (19) accounts for the reset condition (6). The steady-state solution of this system (for constant mean input) can be conveniently calculated[56]. Obtaining the time-varying solution of Eqs. (17–19) is computationally more demanding and can be achieved, e.g., using a finite volume method as described in Supplementary Methods section 2 (see ref. [57]).

As an efficient alternative, reduced models have been developed to approximate the spike rate dynamics of this Fokker–Planck system by a low-dimensional ordinary differential equation (ODE) that can be solved much faster[57–59,98]. Here, we employ a simple yet accurate reduced model from ref. [57] (the LNexp model, based on ref. [58]) adapted for leaky I&F neurons with constant input variance $\sigma^2$. This model is derived via a linear–nonlinear cascade ansatz, where the mean input is first linearly filtered and then passed though a nonlinear function to yield the spike rate. Both components are determined from the Fokker–Planck system and can be conveniently calculated without having to solve Eqs. (17)–(19) forward in time: the linear temporal filter is obtained from the first order spike rate response to small amplitude modulations of the mean input and the nonlinearity is obtained from the steady-state solution[57,58]. The filter is approximated by an exponential function and adapted to the input in order to allow for large deviations of $\mu$. This yields a one-dimensional ODE for the filter application,

$$\frac{d\mu_{\text{f}}}{dt} = \frac{\mu(t) - \mu_{\text{f}}}{\tau_\mu(\mu|\boldsymbol{\theta})}, \qquad (20)$$

where $\mu_{\text{f}}$ is the filtered mean input and $\tau_\mu$ is the (state dependent) time constant. The spike rate is given by the steady-state spike rate of the Fokker–Planck system evaluated at $\mu = \mu_{\text{f}}$,

$$r(t) = r_\infty(\mu_{\text{f}}|\boldsymbol{\theta}). \qquad (21)$$

In order to efficiently simulate this model we pre-calculate $\tau_\mu$ and $r_\infty$ for a reasonable range of mean input values and use look-up tables during time integration. Note that this model is based on the derivation in ref. [58] with filter approximation scheme proposed in ref. [57], which leads to improved accuracy of spike rate reproduction for the sensitive low input regime[57]. For a given mean input time series $\mu[t_0, t]$ we calculate $r(t|\mu[t_0, t], \boldsymbol{\theta})$ using the initial condition $\mu_{\text{f}}(t_0) = \mu(t_0)$.

**Likelihood maximization.** We maximized the logarithm of the likelihood (log-likelihood),

$$\text{argmax}_{\boldsymbol{\theta}} \log p(\text{D}|\boldsymbol{\theta}) = \text{argmax}_{\boldsymbol{\theta}} \sum_{k=1}^{K-1} \log p(t_{k+1}|t_k, \mu[t_k, t_{k+1}], \boldsymbol{\theta}) \qquad (22)$$

for individual neurons, using Eq. (1), and similarly for networks using the logarithm of Eq. (4). Optimization was performed using a simplex algorithm[99] as implemented in the Scipy package for Python. It should be noted that our method is not restricted to this algorithm; alternative, gradient-based optimization techniques, for example, may likely lead to reduced estimation times.

We would further like to remark that maximizing the likelihood $p(\text{D}|\boldsymbol{\theta})$ within plausible limits for the parameter values is equivalent to maximizing the posterior probability density for the parameters given the data, $p(\boldsymbol{\theta}|\text{D})$, without prior knowledge about the parameters except for the limits (i.e., assuming a uniform prior distribution of $\boldsymbol{\theta}$).

**Calculation of the Cramer–Rao bound.** We computed the Cramer–Rao bound for the variance of parameter estimates in Results section "Inference of background inputs". This bound is approached by the variance of a maximum likelihood estimator as the number of realizations increases. Let $\theta$ denote the vector of parameters for estimation (contained in $\boldsymbol{\theta}$), e.g., $\theta = (\mu, \sigma, \tau_m)^T$. In case of a single (non-adapting) model neuron with constant input moments the Cramer–Rao bound for the variance of estimates of $\theta_i$ from spike trains with $K$ spikes is then given by $[\mathcal{I}(\theta)]_{i,i}^{-1}/(K-1)$, where $\mathcal{I}(\theta)$ is the Fisher information matrix per ISI defined by

$$\mathcal{I}(\theta)_{i,j} := -\int_0^\infty p_{\text{ISI}}(s|\mu[0, s], \boldsymbol{\theta}) \frac{\partial^2}{\partial \theta_i \partial \theta_j} \log p_{\text{ISI}}(s|\mu[0, s], \boldsymbol{\theta})ds. \qquad (23)$$

**Modeling input perturbations.** In Results section "Inference of input perturbations" we consider input perturbations of the form $\mu(t) = \mu_0 + J\mu_1(t)$, where $\mu_1(t)$ is described by the superposition of alpha functions with time constant $\tau$, triggered at times $\tilde{t}_1, \dots, \tilde{t}_L$,

$$\mu_1(t) = \sum_{l=1}^{L} H(t - \tilde{t}_l) \frac{t - \tilde{t}_l}{\tau} \exp\left(1 - \frac{t - \tilde{t}_l}{\tau}\right), \qquad (24)$$

with Heaviside step function $H$. The alpha functions are normalized such that their maximum value is 1 when considered in isolation. As an alternative we also considered delayed delta pulses instead of alpha kernels

$$\mu_1(t) = \sum_{l=1}^{L} \delta(t - \tilde{t}_l - d), \qquad (25)$$

where $d$ denotes the time delay. The perturbation onset (trigger) times were generated by randomly sampling successive separation intervals $\tilde{t}_{l+1} - \tilde{t}_l$ from a Gaussian distribution with 200 ms mean and 50 ms standard deviation.

In Fig. 3 and Supplementary Fig. 3, we quantified the sensitivity to detect weak input perturbations using our estimation methods (1a and 2) in comparison with a detection method based on the generated data only. For a given parametrization $N_r$ spike trains were simulated using different realizations of neuronal noise and perturbation onset times. Detection sensitivity was quantified by comparing the inferred perturbation strengths from data generated with ($J \neq 0$) and without ($J = 0$) perturbations.

For the I&F-based methods it was calculated by the fraction of $N_r = 50$ estimates of $J$ for true $J > 0$ ($J < 0$) that exceeded the 95th percentile (fell below the 5th percentile) of estimates without perturbation. The model-free reference method was based on CCGs between the spike trains and perturbation times (in other words, spike density curves aligned to perturbation onset times). For each realization one such curve was calculated by the differences between spike times and the perturbation onset times using a Gaussian kernel with 3 ms standard deviation. Detection sensitivity was assessed by the fraction of $N_r = 300$ CCGs for which a significant peak (for $J > 0$) or trough (for $J < 0$) appeared in the interval [0, 100] ms]. Significance was achieved for true $J > 0$ ($J < 0$) if the curve maximum (minimum) exceeded the 95th percentile (fell below the 5th percentile) of maxima (minima) in that interval without perturbation.

**Network model and inference details.** In Results section "Inference of synaptic coupling" we consider networks of $N_{tot}$ coupled leaky I&F neurons from which the spike trains of $N \leq N_{tot}$ neurons have been observed. These networks are given by

$$\frac{dV_i}{dt} = -\frac{V_i}{\tau_m} + \mu_i(t) + \sum_{j=1}^{N_{tot}} J_{i,j}\mu_j^1(t) + \sigma_i\eta_i(t), \tag{26}$$

$$\mu_i^1(t) = \sum_{k=1}^{K_i} \delta(t - t_i^k - d_{i,j}), \tag{27}$$

$$\text{if } V_i(t) \geq V_s \text{ then } V_i(t) \leftarrow V_r, \tag{28}$$

for $i \in \{1, \ldots, N_{tot}\}$, where $J_{i,j}$ denotes the coupling strength between presynaptic neuron $j$ and postsynaptic neuron $i$, $t_i^k$ is the $k$th of $K_i$ spike times of neuron $i$, and $d_{i,j}$ is the delay. $\eta_i$ describes the fluctuations of external input received from unobserved neurons,

$$\eta_i(t) = \sqrt{1-c}\,\xi_i(t) + \sqrt{c}\,\xi_c(t), \tag{29}$$

where $\xi_i, \xi_c$ are independent unit Gaussian white noise processes, i.e., $\langle\xi_i(t)\xi_j(t+\tau)\rangle = \delta_{ij}\delta(\tau)$, $i, j \in \{1, \ldots, N_{tot}, c\}$, and $c$ is the input correlation coefficient. We considered uncorrelated or weakly correlated external input fluctuations, i.e., $c = 0$ or $c = 0.1$. Note that the input variation for neuron $i$ caused by (observed or unobserved) neuron $j$ across the interval $[t_i^k, t_i^{k+1}]$, denoted by $J_{i,j}\mu_j^1[t_i^k, t_i^{k+1}]$ (as used in Eq. (4)), is determined by the spike times of neuron $j$ that occur in the interval $[t_i^k - d_{i,j}, t_i^{k+1} - d_{i,j}]$. For simulated data we chose identical delays across the network, but this is not a restriction of our inference method (see below). Coupling strengths were uniformly sampled in [−0.75, 0.75] mV (Fig. 5a, b and Supplementary Fig. 5a), otherwise excitatory/inhibitory connections were randomly generated with specified probabilities and coupling strengths were then uniformly sampled with mean ±0.5 mV (Fig. 5c–k and Supplementary Fig. 5b–f) or mean ±1 mV (Supplementary Fig. 6), respectively. Autapses were excluded, i.e., $J_{i,i} = 0$. Network simulations for Fig. 5c–k, Supplementary Figs. 5b–f and 6 were performed using the Python-based Brian2 simulator[100].

Our method fits an I&F network, described by Eqs. (26)–(29) with $c = 0$ for the $N$ observed neurons (i.e., $i \in \{1, \ldots, N\}$) to the spike train data by maximizing the likelihood (4). In the fitted model the effects of unobserved neurons are absorbed by the parameters $\mu_i(t)$ and $\sigma_i$, where $\mu_i(t)$ is discretized in an event-based way: the background mean input for neuron $i$ during its $k$th ISI is represented by a constant value $\mu_i^k$ (cf. Eq. (4)). In Fig. 5, Supplementary Figs. 5 and 6 $\mu_i(t)$ was assumed to be constant over time, whereas in Fig. 6 and Supplementary Fig. 7 it varied between ISIs; for details on the estimation of these variations see Methods section "In vivo ground truth data on synaptic connections".

The logarithm of the spike train likelihood (4) (cf. Methods section "Likelihood maximization") was optimized in the following way, justified by the assumption of weak coupling. First, the parameters of the background input, $\mu_i^k$ and $\sigma_i$, were estimated for each neuron in isolation (all $J_{i,j} = 0$). Consequently, effects of observed and unobserved neurons are reflected by these estimates. Then, the coupling strength $J_{i,j}$ and delay $d_{i,j}$ were estimated given $\mu_i^k$ and $\sigma_i$ for each $i,j$-pair. These two steps were performed in parallel over (postsynaptic) neurons. We estimated couplings in a pairwise manner to save computation time, assuming that the transient effects of an individual synaptic connection on the spiking probability of a neuron are negligible for the estimation of other synaptic connections. This is justified for weak coupling and network activity where synchronous spikes across the network occur sparsely.

We then corrected for a potential systematic bias in the estimated coupling strengths of a network in Fig. 5, Supplementary Figs. 5 and 6 as follows. We perturbed all presynaptic spike times by a temporal jitter (random values uniformly sampled in the interval [−10, 10] ms) to mask any of the transient effects caused by synaptic connections, and re-estimated the coupling strengths for multiple such realizations for each postsynaptic neuron $i$ given $\mu_i^k$ and $\sigma_i$. The averaged bias that was estimated across the network from this procedure was then subtracted from the original estimates.

For comparison we used a method that fits a point process GLM to the data. From the class of GLM-based approaches[2,3,80] we chose one that is well suited for reconstruction from spike train data generated by an I&F network as specified above and for which an efficient implementation is available[66]. In the GLM network model the incoming spike trains, after incurring transmission delays, are filtered by a leaky integrator with a time constant and a (constant) baseline activity parameter for each neuron. The resulting membrane potential is passed through an exponential link function, which transforms it into the time-varying rate of a Poisson point process that generates the output spike train of the neuron. The spike train is also fed back as an input to the neuron itself to model refractory, post-spike properties. The coupling terms in the GLM and I&F networks are equivalent: both models use delayed delta pulses.

Using maximum likelihood estimation this GLM method inferred $N^2 + 3N$ parameters per network with $N$ neurons: the coupling strengths (including self-feedback) as well as time constant, baseline parameter and delay, one for each neuron. Note that for the simulated data only one (global) delay value was used for all connections in a network. Hence, the GLM method estimated fewer parameters compared to the I&F method (which inferred $2N^2$ parameters). For details on the elaborate inference technique, which includes regularized optimization and cross-validation, and an available Python implementation using the library Cython for accelerated program execution we refer to ref. [66].

In the model-free, CCG-based method a connection strength was estimated by the $z$-score of the extremum in the spike train CCG across positive lags for each pair of neurons. Note that the lag denotes the time since a presynaptic spike. $z$-scores were obtained using estimates from surrogate data generated by perturbing the presynaptic spike times by a temporal jitter (random values between −10 ms and 10 ms) in a large number of realizations.

For each of the three methods detailed in this section we assessed detection performance in the following way. A discrimination threshold $J_{thresh}$ for the presence or absence of connections was applied to estimated coupling strengths $\hat{J}_{i,j}$. Accordingly, the presence of a connection ($i,j$-pair) was assured by the condition $|\hat{J}_{i,j}| > J_{thresh}$. The true positive rate (sensitivity) was given by the number TP of connections for which the estimation satisfied $|\hat{J}_{i,j}| > J_{thresh}$ and a true connection was present ($|J_{i,j}| > 0$), divided by the number P of true connections. The true negative rate (specificity) was given by the number TN of connections for which $|\hat{J}_{i,j}| \leq J_{thresh}$ and a true connection was absent ($J_{i,j} = 0$), divided by the number N of absent connections. Receiver operating characteristic (ROC) curves were generated from sensitivity and specificity as a function of $J_{thresh}$. Accuracy (ACC) and balanced accuracy (BACC) are defined as ACC = (TP + TN)/(P + N) and BACC = (TP/P + TN/N)/2, respectively.

**Modeling spike rate adaptation.** In Results section "Inference of neuronal adaptation" we consider an extended I&F neuron model that includes an additional adaptation (current) variable $w$ that is incremented at spike times, slowly decays, and counteracts the input to the neuron[14,55]; Eqs. (5) and (6) where the mean input $\mu(t)$ is replaced by an effective mean input $\mu(t) - w(t)$, with

$$\frac{dw}{dt} = -\frac{w}{\tau_w}, \tag{30}$$

$$\text{if } V(t) \geq V_s \text{ then } w(t) \leftarrow w(t) + \Delta w. \tag{31}$$

Here, $\tau_w$ is the adaptation time constant and $\Delta w$ denotes the spike-triggered increment.

For known spike times, contained in set D, the effective mean input can be written as $\mu(t) - \Delta w\mu_1(t|D, \tau_w)$, where $\mu_1$ between spike times $t_k$ and $t_{k+1}$ is explicitly expressed by

$$\mu_1(t|D, \tau_w) = \sum_{i=1}^{k} H(t - t_i)\exp\left(-\frac{t - t_i}{\tau_w}\right), \tag{32}$$

$t \in [t_k, t_{k+1}]$, with Heaviside step function $H$, assuming the adaptation current just prior to the first spike is zero, $w(t_1^-) = 0$. This means, for given parameters $\mu$, $\Delta w$, $\tau_w$ the effective mean input time series is determined by the observed spike train (up to $t_k$).

Note that in general the mean input perturbations caused by adaptation vary from spike to spike, $\mu_1(t_k|D, \tau_w) \neq \mu_1(t_l|D,\tau_w)$ for $t_k \neq t_l \in D$. To efficiently evaluate the likelihood $p(D|\theta)$ via method 1 (using Eqs. (1) and (8)) we calculate $p_{ISI}(s|\mu_{ISI}[0, s],\theta)$ with $\mu_{ISI}(s) = \mu(s) - w_0 \exp(-s/\tau_w)$, $s \geq 0$ for a reasonable range of values for $w_0$ and interpolate to obtain $p_{ISI}(s_k|\mu_{ISI}[0, s_k],\theta)$ with $\mu_{ISI}[0, s_k] = \mu[t_k, t_{k+1}] - \Delta w\,\mu_1[t_k, t_{k+1}]$ using Eq. (32). Methods 1a and 2 are less well suited for this scenario because the adaptation variable can accumulate to substantial values, thereby opposing the

assumption of weak variations of the mean input; moreover, the spike train of an adapting neuron deviates strongly from a Poisson process.

**Implementation and computational complexity**. We have implemented our methods for parameter estimation (1, 1a, and 2) using the Python programming language and applying the libraries Scipy[101] for optimization and Numba[102] for low-level machine acceleration. The code for representative estimation examples from Results sections "Inference of background inputs" to "Inference of neuronal adaptation" is available at GitHub: https://github.com/neuromethods/inference-for-integrate-and-fire-models. Computation times for example inference problems are summarized in Supplementary Table 1.

**In vitro ground truth data on neuronal input statistics**. We used somatic whole-cell current clamp recordings from primary somatosensory cortex in acute brain slices (for details see ref. [49]). Layer 5 PYRs were recorded in wild-type mice[49], fast-spiking layer 5 INTs were selected among the fluorescing cells of a GAD67-GFP transgenic line[103]. Only cells with an access resistance ≤25 MΩ (PYR: 18.3 ± 1.5 MΩ, $n = 7$; INT: 19.5 ± 4.0 MΩ, $n = 6$) and a drift in the resting membrane potential ≤7.5 mV (PYR: 3.2 ± 3.0 mV, $n = 7$; INT: 3.1 ± 3.7 mV, $n = 6$) throughout the recording were retained for further analysis. Seven PYRs and six INTs were stimulated with a fluctuating current $I(t)$ generated according to an Ornstein–Uhlenbeck process

$$\frac{dI}{dt} = \frac{\mu_I - I}{\tau_I} + \sqrt{\frac{2}{\tau_I}}\sigma_I \xi(t), \qquad (33)$$

where $\tau_I$ denotes the correlation time, $\mu_I$ and $\sigma_I$ are the mean and standard deviation of the stationary normal distribution, i.e., $\lim_{t\to\infty} I(t) \sim \mathcal{N}(\mu_I, \sigma_I^2)$, and $\xi$ is a unit Gaussian white noise process. Somatic current injections lasted 5 s and were separated by inter-stimulus intervals of at least 25 s. Different values for $\mu_I$ and $\sigma_I$ were used and each combination was repeated three times. The correlation time was set to 3 ms. Spike times were defined by the time at which the membrane voltage crossed 0 mV from below, which was consistent with a large depolarization rate $dV/dt > 10$ mV/ms[49]. An absolute refractory period of 3 ms was assumed.

For each neuron we fitted a leaky I&F model with and without adaptation (cf. Methods sections "I&F neuron models" and "Modeling spike rate adaptation"). Note that the injected current $I(t)$ can be well approximated by a Gaussian white noise process as considered in our model because of the small correlation time $\tau_I$. In Results section "Inference of background inputs" we estimated the input parameters $\mu$ and $\sigma$ for nonadaptive model neurons from each 5-s long spike train recording as well as from each combined $3 \times 5$-s long recording (using the three repetitions with identical stimulus parameters which effectively yielded 15 s long stimuli). To exclude onset transients (i.e., increased spike rate upon stimulus onset) we used the central 90% of ISIs for each stimulus, ensuring that ISIs lasted >5 ms. For comparison we considered a Poisson process with constant rate. In Results section "Inference of neuronal adaptation" we additionally estimated the adaptation parameters $\Delta w$ and $\tau_w$ per neuron across all available stimuli in the combined 15 s stimulus window. Here we used all ISIs (including the short ones at stimulus onset) in order to unmask adaptation effects. Parameter estimation was accomplished using method 1. To compare the quality of the models and avoid over-fitting we used the AIC[53,104], given by $2N_\theta - 2\max_\theta \log p(D|\theta)$, where $N_\theta$ denotes the number of estimated parameters ($\theta$ is a subvector of $\theta$) for a particular model. For the adaptive I&F model $N_\theta = 4$, for the nonadaptive I&F model $N_\theta = 2$, and for the Poisson model $N_\theta = 1$. The preferred model from a set of candidate models is the one with the smallest AIC value.

**Estimating neuronal input statistics from in vivo data**. We used single unit spike trains from extracellular recordings of two adult female ferrets in an awake, spontaneous state. The animals were listening to acoustic stimuli separated by periods of silence lasting 0.4 s which we used for model fitting. Neural activity from primary auditory cortex was recorded using a 24 channel-electrode and spikes were sorted using an automatic clustering algorithm followed by a manual adjustment of the clusters (for details see ref. [52]). Spike trains with >50 ISIs during silence periods were considered for fitting. Seventy-one single units passed that threshold in each of two behavioral conditions (passive listening vs. engaged in a discrimination task). Model neurons were fit in either behavioral condition separately, resulting in 142 sets of estimated parameters. We employed the leaky I&F model (Eqs. (5) and (6)) with constant background input mean $\mu$. For robust estimation we used the central 95% of ISIs, ensuring that ISIs lasted >2.5 ms. For comparison we considered a Poisson process with constant rate and compared the quality of the models using the AIC.

**In vitro ground truth data on input perturbations**. We used whole-cell recordings of pyramidal neurons in slices of rat visual cortex. Ten neurons were stimulated with an input current that consisted of transient bumps reflecting an aEPSC immersed in background noise (for details see ref. [50]: experiment 1). The background noise was generated as in Methods section "In vitro ground truth data on neuronal input statistics" with $\tau_I = 5$ ms, $\mu_I$ tuned to maintain ~5 spikes/

s, and $\sigma_I$ adjusted to produce membrane voltage fluctuations with ~15–20 mV peak-to-peak amplitude. aEPSC traces were generated by convolving a simulated presynaptic spike train with a synaptic kernel described by the difference of two exponentials (rise time 1 ms, decay time 10 ms). Presynaptic spikes were generated by a gamma renewal process (shape 2, scale 2.5) with 5 spikes/s on average; aEPSC amplitudes triggered by a single spike ranged from 0.1 $\sigma_I$ to 1.5 $\sigma_I$. Current was injected in segments of 46 s length with at least 10 repetitions per aEPSC strength (except for one cell). The first and last 3 s of each segment were discarded from the analysis, as in ref. [50].

Using only the presynaptic and postsynaptic spike times, we fitted an I&F neuron where input perturbations were described using delta pulses or alpha functions (cf. Methods section "Modeling input perturbations"). For the former model we applied method 1a, for the latter method 2. Similarly to ref. [50] we defined detection time as the minimal total length of spike train data for which the model with input perturbations ($J \neq 0$) yields a larger likelihood on test data compared to the respective model without input perturbations ($J = 0$); this was indicated by a positive log-likelihood ratio on test data from 10-fold cross-validation. In addition, we assessed detection performance on a fixed amount of data that consisted of five consecutive segments (i.e., 200 s recording duration), where adjacent five-segment blocks shared one segment. Coupling strength $z$-scores were computed using estimates from surrogate data generated by perturbing the presynaptic spike times by a temporal jitter (cf. Methods section "Network model and inference details") in a large number of realizations.

**In vivo ground truth data on synaptic connections**. We used combined juxtacellular–extracellular recordings of neuronal ensembles from the hippocampal CA1 region in awake mice (for details see ref. [51]). Neurons were separated into PYRs and INTs according to their spike waveform and spiking statistics. Spikes were evoked in single PYRs by short current pulses (50–100 ms) applied at intervals of variable length using juxtacellular electrodes while recording extracellular spikes of local INTs. PYR spikes which occurred during a stimulus were considered as evoked, and those which occurred at all other times were considered as spontaneous. All spikes that occurred during sharp-wave ripple events were discarded from the analyses and we only considered INTs that fired at least 3 spikes/s on average. A total of 78 PYR-INT pairs were included for estimation of synaptic couplings.

For each INT we fitted a leaky I&F neuron receiving background input and (potential) synaptic input from the recorded PYR such that each presynaptic spike causes a delayed postsynaptic potential with delay $d$ and size $J$ (cf. Eqs. (26–29) with $c = 0$, where we omit the indices $i, j$ here for simplicity).

To account for changes in background input statistics over the recording duration, which lasted up to ~2 h, and to reflect low-frequency network co-modulation induced by common network drive, the background mean input was allowed to vary over time. The parameters to be estimated are thus $\mu(t)$, $J$, $d$, and $\sigma$. Estimation consisted of three steps. First, we inferred the statistics $\mu(t)$ and $\sigma$ of background inputs for $J = 0$ in the following way. We computed the empirical instantaneous spike rate $r(t)$ of the INT from the observed spike train via kernel density estimation using a Gaussian kernel with width $\sigma_G \in \{0.1, 0.5, 1\}$ s. The estimated empirical spike rate varies over time much slower than the timescale at which changes of mean input translate to changes of spike rate in the I&F model. This justifies the approximation $r(t) \approx r_\infty(\mu(t)|\theta)$ (cf. Methods section "Method 2: derived spike rate model"), which allowed us to efficiently evaluate the spike train likelihood for fixed $\sigma$ by applying method 1 with mean input assumed constant within each ISI, given by $\mu(t) = r_\infty^{-1}(r(t)|\theta)$ (at the center between consecutive spike times). The likelihood was then maximized with respect to $\sigma$. Given the parameters for the background inputs (one value of $\mu$ per ISI and one for $\sigma$) we next maximized the likelihood of the full model in the second step with respect to $J$ and $d$ using method 1a. In the third step we assessed the significance of synaptic coupling estimates using surrogate data, similarly as in the previous section. We perturbed the presynaptic spike times by a small temporal jitter (random values between −5 and +5 ms) and re-estimated $J$ and $d$. This was repeated 100 times and $z$-scores were computed from the estimated coupling strengths. Notably, since spike times are shifted by only small values, effects due to network co-modulation which occurs on a slower timescale are preserved in the surrogate data. In this way we obtained a coupling strength $z$-score for each PYR-INT pair and for each of the three values of $\sigma_G$.

We validated our results against ground truth connection labels obtained from juxtacellular evoked activity using a model-free method based on spike train CCGs[51] (for details see Supplementary Methods section 4). Based on these labels we computed ROC curves as well as ACC and BACC (cf. Methods section "Network model and inference details") using a classification ($z$-score) threshold value $J_{\text{thresh}}^z$. Accordingly, the presence of an estimated connection was assured by the condition $\hat{J}^z > J_{\text{thresh}}^z$, where $\hat{J}^z$ denotes the connection strength ($z$-score) estimate for a given PYR-INT pair. Note that the ground truth labels indicate excitatory connections (positives) and absent connections (negatives).

To test the validity of our approach we estimated connectivity using only the first evoked PYR spikes of each stimulation pulse, which are maximally decoupled from network co-modulation, and compared the results with the ground truth labels. This assessment yielded excellent agreement, with ACC and BACC values of up to 0.97 and 0.95, respectively (for $\sigma_G = 0.1$ s).

**Reporting summary**. Further information on research design is available in the Nature Research Reporting Summary linked to this article.

## Data availability

No experimental data were collected for this study. The study involved available datasets from previous experimental studies with ethical approval granted. These datasets are available either online https://doi.org/10.6084/m9.figshare.1144467 or from the authors on reasonable request.

## Code availability

Python code for our methods and estimation examples are available under a free license at https://github.com/neuromethods/inference-for-integrate-and-fire-models.

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

## Acknowledgements

We thank Dimitra Maoutsa for her support on the GLM implementation. This work was supported by Deutsche Forschungsgemeinschaft in the framework of Collaborative Research Center 910, the Programme Emergences of the City of Paris, ANR project MORSE (ANR-16-CE37-0016), and the program "Ecoles Universitaires de Recherche" launched by the French Government and implemented by the ANR, with the reference ANR-17-EURE-0017. The funders had no role in study design, data collection and analysis, decision to publish, or preparation of the paper.

## Author contributions

J.L. and S.O. designed the study. J.L. led the project, developed and implemented the methods, performed the evaluations, validations and benchmark tests, and wrote the paper. S.M., D.E., and O.H. provided electrophysiological data. S.O. supervised the study and edited the paper. S.M. and O.H. provided feedback on the paper.

## Competing interests

The authors declare no competing interests.
