## [Peer Review File · Nature Communications]

Reviewers' Comments:

Reviewer #1:

Remarks to the Author:

The authors develop a set of techniques for fitting integrate-and-fire neuron models to data, using a maximum-likelihood approach based on spike times. The techniques involve solving or approximating a Fokker-Planck equation to evaluate inter-spike-interval likelihoods. The techniques are validated using simulated networks and data from mouse hippocampal neurons for which ground-truth connectivity data is available. Overall, I think this is a valuable application of these Fokker-Planck approaches to an inference problem, and suggests more principled ways of building integrate-and-fire models to match to data. The paper is well-written and clear, so I have only a few comments.

Comments:

1) In Section 2, why was the membrane time constant set to the true value? Could it not also be estimated?

2) I would like to see an expanded discussion of the issue of non-stationarity (page 10) in the main text. This seems like it would be a major issue for any in vivo recording. Currently, the text mentions two methods and an analysis of variations of mean input at three different timescales, but the information provided is insufficient to understand what actually is being performed. The Methods section is more helpful, but could use additional justification of the methods (why three timescales, and what is the justification for the "combined timescale" estimate?).

3) I was unclear why the input standard deviation in Fig. 7 wouldn't change between models with and without adaptation. Is there a reason that the two parameters should be completely decoupled from one another in the inference? This should be explained more fully.

Reviewer #2:

Remarks to the Author:

The authors develop maximum likelihood based approaches to fitting integrate-and-fire models to extracellular recordings, specifically approaches that allow to use data that consists solely of spike times. Since population extracellular recordings are far more common than access to the membrane voltage of a population of neurons, there is value in such approaches.

My main concern is that the paper addresses a problem that has been repeatedly considered in prior work, yet is unclear about comparison to existing techniques. The paper considers multiple inference scenarios and I am still unclear as to in which (if any) of these scenarios does the approach allow biologically interesting information to be learned in a qualitatively superior way to that possible in previous approaches. I would also be interested if that is not the case, and the approach is just more computationally efficient, quantitatively more accurate, can be used with less data, or perhaps just more principled. But whichever of these claims are true they need to be clearly described and quantified for each scenario considered. As the paper stands I found this information too difficult to parse out and cannot recommend publication in its current format.

Major comments:

1. As stated above, my main concern is that I found it extremely difficult to parse out what specifically this approach enables that cannot be done with previous methods, or the specific advantages over existing methods. The paper considers multiple scenarios:
 - i. Mean and standard deviation of input to a single neuron.

- a. Synthetic data
- b. In-vitro
- ii. EPSP shaped perturbations
- iii. Synaptic inference
 - a. Simulation fully observed
 - b. Simulation partially observed
- c. In-vivo
- iv. I&F models with adaptation
- v. Estimation of neuronal inputs from in-vivo recordings

The authors did include quantitative comparison to existing techniques in the results or figures. Some of these, for instance item i, have been directly addressed by previous approaches and many of these could have easily been compared to a variety of previous approaches, either more principled approaches or more straightforward approaches. The authors often report high accuracy for their results, but such numbers are very difficult to interpret as absolute numbers without reference to a comparison value. For each item the authors must either present the most pertinent comparison, or ideally comparisons to multiple existing techniques, while including straightforward extensions where necessary (e.g., grid searches for simple unknown but necessary parameters such as mean input current). Then the authors can clearly layout what specifically their approach improves: accuracy, computation time, amount of data necessary, etc. Where there is no substantial improvement, as I would expect to be the case for instance in item i, the authors can either present this as a useful introductory analysis, or claim that they prefer their technique since it is more principled, or something else. But this has to be much clearly and explicitly presented per result and not just in a somewhat vague Discussion paragraph.

2. Related to this point, Figure 5 which discusses inference of synaptic connectivity on experimental data, appears to describe the results that are closest to what the paper aims to do, or at least that is what I understand inferring microcircuits, and the data described come with associated semi-ground truth. Therefore the lack of comparison here is most frustrating. The ROC curves look very good, which seems impressive, yet the empirical CCG presented shown in bottom left of Fig. 5A also looks very different from background, leading one to assume that straightforward techniques would also work quite well. As a matter of fact, reference 35, from which the authors took the data, also claims accurate inference of connections from CCG itself. Perhaps the current approach does better, but explicit comparison has to be made to allow a reader to appreciate that.

3. I couldn't understand the results related to subsampling observed neurons from a larger simulated network, which is the more relevant regime to look at for simulated data. The authors simply state that: "Estimation accuracy of connectivity structure and coupling strengths was surprisingly good for sufficiently long simulated recordings in this scenario.". This is indeed surprising and requires more clarification.

4. The logic of section 6 seemed strange to me. The main point seems to be Figure 8E-F, the balance of excitation and inhibition inferred by their model, but the authors go through a path of claiming that their models are effective that is unclear and not entirely convincing to me. The authors assess their approach by comparing it to simpler models, but these models appear to be rather weak models: "To assess whether the complexity of this model is adequate given the available data we considered simpler models for comparison. For spontaneous activity, we compared our model to a Poisson process with constant rate. For click-evoked activity, we examined two additional models: (i) the leaky I&F model with only one (either excitatory or inhibitory) click-triggered input described by a delayed alpha function; (ii) a model in which the stimulus induced only a constant additional input." Each of these comparisons takes out a key component that is very likely to be important and therefore neither of these comparisons seem very informative. It was difficult for me therefore to judge the

reasonableness of Figure 8E-F. The authors make strong claims regarding this point: "Fitting only spike-trains, our approach therefore uncovers fundamental constraints on synaptic inputs and the local microcircuit without access to the values of the intra-cellular membrane potential." Accordingly, more support and explanation would be helpful.

5. Though this comment is subjective, I found the paper's tendency to oversell its results unappealing. Specifically, the paper pushes some of the main terminology beyond their natural usage. First, the authors refer to I&F models as mechanistic models. This is simply not the case, it is a phenomenological model. I refer the authors to the first paragraph of Abbott (1999) on the I&F model: "In 1907, long before the mechanisms responsible for the generation of neuronal action potentials were known, Lapicque developed a neuron model that is still widely used today [3,7]. This remarkable achievement stresses that, in neural modeling, studies of function do not necessarily require an understanding of mechanism. Significant progress is possible if a phenomenon is adequately described, even if its biophysical basis cannot be modeled.". Moreover multiple other papers referenced by the authors (if not all of them) properly refer to I&F models as phenomenological models. I&F models can certainly be viewed as closer to mechanism than abstract statistical models, as the authors claim, but this does not justify calling it a mechanistic model. Similarly, the paper claims to be "Inferring and validating mechanistic models of neural microcircuits based on spike-train data" but most of the results are inference of inputs to single neurons, which is not the standard usage for microcircuit models (though admittedly this is a not well defined term). It appears that the authors feel that single I&F models can be referred to as a microcircuit model: "We fitted a simple microcircuit model to these data. In the model, a leaky I&F model neuron received feed-forward excitatory and inhibitory inputs triggered by the clicks, as well as a fluctuating background input." But this is hardly what comes to my mind when one refers to a microcircuit model. I understand the desire to make the paper more likely to be read by a general audience, but I feel the authors would be better served by not stretching definitions.

Minor comments:

1. In general I found multiple figures to be crowded and ineffective. There is often a lot of detail in the figure that is very difficult to know how to interpret while crucial detail remains difficult to judge. For instance, Figure 8A bottom appears to contain crucial information regarding the comparison that will ultimately be made in Figure 8E-F, but it is tiny, not well-described and overwhelmed by the much larger (and seemingly less informative) subplots 8B-D.

2. Often important information regarding the results is present only in the methods, and sometimes sparsely described there, for instance the sensitivity measure used in Figure 3D.

3. Percentile is sometimes written as "%-ile". This is non-standard usage and should be avoided.

4. Section regarding figure 4: "The estimation accuracy of synaptic strengths is still good in case of weak correlations of the external input fluctuations (correlation coefficient $c \leq 0.05$ for each pair of observed neurons) but clearly decreases as these correlations increase (Fig. 4E). In particular, positive correlations $c > 0$ lead to an overestimation of the coupling strengths."

Figure 4E shows only two scatters, one for $c=0.05$ and one for $c=0.1$. The Pearson's correlation of the former is 0.95 and the latter 0.94. The described shift up of weights can be observed but is not quantified, and the quantification doesn't show a decrease in accuracy, which makes relating the figure to the text confusing.

5. Top of page 11: "for the combined timescales variant; when using all instead of max. thresh 1000 PYR spikes....". Max. as an abbreviation should be avoided in the text.

6. Page 11 describing adaptation: "This behavior is typically mediated by a slowly decaying transmembrane potassium current, which rapidly accumulates when the neuron spikes repeatedly." I couldn't tell with certainty if this is meant as an abbreviated description of calcium dependent potassium currents or not. The authors don't have to mention this, but if they do a clearer description (including a reference) would be useful.

7. As stated above I wouldn't call section 6 fitting a simple microcircuit model.

8. The authors sometimes include words in parentheses for unclear reasons. For instance: "This enables to better isolate pairwise interactions from (ubiquitous) fluctuating drive..." either ubiquitous is an important word in this center and should be retained without parentheses or it can be removed. Unclear usage of parentheses can be confusing.

9. "Neurons were separated into PYRs and INTs according to their spiking statistics." The term spiking statistics could be read to indicate that spike waveform was not used, which is not the case, at least in the paper that the authors reference.

Reviewer #3:

Remarks to the Author:

In this manuscript, Ladenbauer and colleagues present a probabilistic approach to fit integrate and fire models to extracellularly recorded neurons. The methods are used to infer connectivity between neurons and stimulus-response relationships. This method aims to bridge the gap between purely phenomenological tools and biophysical descriptions of neurons.

Unfortunately, I have strong concerns about the novelty of the methods and the strength of the claims that the LIF approach improves upon purely statistical models. Thus, while the general goals behind this methods paper are of wide interest, I cannot recommend this paper to be published in Nature Communications at this time.

My main critique of this paper concerns the novelty of the integrate-and-fire methodology. Several of the modeling problems presented in the results section fall within the generalized integrate and fire model from Pillow et al. (2005), for which code is publicly available. Comparing the results from Ladenbauer and colleagues to equation 1 from Pillow et al:

- results section 2 considered the case where $I_{\text{stim}} = \text{constant}, I_{\text{spk}} = 0$.
- results section 3 uses the model with $I_{\text{spk}} = 0$ the stimulus is μ_1
- results section 5 includes the I_{spk} term which captures adaptation
- results section 6 applies the full generalized IF model to spiking data in response to a sensory stimulus, as Pillow et al did with retinal ganglion cells (but with a specific decomposition of the linear stimulus filter).

Given previous work by Paninski and colleagues cited in this manuscript that established the log concavity of the generalized IF's likelihood function, accurately recovering the IF model's parameters with maximum likelihood estimation given enough data in the simulation studies here was expected.

One of the major contributions of this paper is estimating connectivity between extracellularly recorded neurons. The discussion states that the IF model can "better isolate pairwise interactions from (ubiquitous) fluctuating drive". The introduction claims that statistical models suffer from a lack of connection to biophysical structures and that this paper serves to bridge the gap between biophysical and statistical models. However, the authors have not cited an extremely similar study using generalized linear models (GLMs) to tackle the problems in results sections 3-4 (while also

including adaptation): Volgushev et al. (2015). Thus, these claims are not substantiated because the LIF model isn't compared to anything like GLMs to show that the biophysical approach shows more than can be seen with a phenomenological model.

I would still encourage the authors to build on the novel content of this paper for future publications. This includes:

1. Faster estimation of the stochastic LIF model. How do methods 1a,b and 2 compare in speed and accuracy to previous Fokker-Planck estimation tools? The code from Pillow et al is available from the Pillow lab website for comparison.
2. Application of the generalized LIF to ferret cortex. What do we learn from this model fit?
3. LIF for estimating connectivity. How does this method compare to the GLM estimator in Volgushev et al? Does this biophysical tool validate the more abstract GLM approach? Or does the biophysical model pick up on things the GLM misses? Data, but not code, from the Volgushev paper is available from the Stevenson lab website.

References:

Pillow JW, Paninski L, Uzzell VJ, Simoncelli EP, Chichilnisky EJ. (2005). Prediction and Decoding of Retinal Ganglion Cell Responses with a Probabilistic Spiking Model. *Journal of Neuroscience* 25:11003-11013.

Volgushev M, Ilin V, Stevenson IH (2015). Identifying and Tracking Simulated Synaptic Inputs from Neuronal Firing: Insights from In Vitro Experiments. *PLoS Comput Biol* 11(3): e1004167.doi:10.1371/journal.pcbi.1004167

Specific comments and suggestions:

- The presentation of statistical analyses was difficult to follow. Population and perturbation results (Figs 3-5) use the likelihood formulation for fitting the IF models, but do not appear to take advantage of the probabilistic formulation for model selection/hypothesis testing (does a connection exist? Is $J > 0$?). For example, I found the methods used to calculate detection sensitivity in Figure 3D unclear and difficult to compare between CCG and LIF modeling. It was notable for the adaptation/in vivo results (results sections 5 & 6) changed to use AIC for model selection; the paper would benefit from applying this approach more consistently.

- Equations 15 & 50 show that the coupling inputs are limited to delta functions (in contrast to Volgushev et al). Bottom left of Fig 5A suggests that the delta shape misses prediction of the true CCG. Does this approximation limit/bias detection of more realistic EPSP shapes?

- Figure 4: C-D – Estimated inhibitory coupling weights appear biased towards 0. Are excitatory weights estimated more accurately?

E - Is the effect of correlation a constant positive bias? How does this compare with shuffling-based CCG methods for detecting pairwise correlations?

- In section 6, the excitatory and inhibitory components of the stimulus response are additive. Is this decomposition of input into these components unique? And can they be compared to recorded synaptic currents (if available)?

- Section 4.2 and Fig 5A – The method for estimating the mean rate appears ad hoc, and is not justified in simulation. Page 10 indicates that 3 timescales were used to estimate the mean rate and

the results were combined "using the largest absolute z-score across the three timescales for each connection." I am unsure what this means.

Response to the reviewers

We thank the reviewers for their constructive criticism and helpful suggestions, which led us to substantially extend our study and the manuscript. We modified all main figures, and added 9 supplementary ones to address all the suggestions. In particular, we

- 1) included systematic comparisons with related methods,
- 2) significantly extended and advanced our results on inference of synaptic couplings in sub-sampled networks,
- 3) incorporated an additional ground truth dataset,
- 4) clarified the benefits of our approach throughout the paper.

Our comprehensively revised manuscript now includes an extensive number of validations, systematically using ground truth data from in-vitro and in-vivo electrophysiological recordings for all considered inference scenarios (four different datasets), and benchmarks against several existing model-based and model-free approaches.

We would further like to emphasize the novelty of our methods for the inference of connectivity in spiking network models in an in-vivo setting; related approaches have been previously applied for single I&F neurons (Pillow et al. 2005, Paninski et al. 2004) or networks limited to vanishing input fluctuations (Cocco et al. 2009). Instead here we focus on strongly fluctuating inputs that are required to produce in-vivo-like spiking statistics.

Below we provide detailed responses to the reviewers' comments (comments in blue, answers in black).

Reviewer #1 (Remarks to the Author):

The authors develop a set of techniques for fitting integrate-and-fire neuron models to data, using a maximum-likelihood approach based on spike times. The techniques involve solving or approximating a Fokker-Planck equation to evaluate inter-spike-interval likelihoods. The techniques are validated using simulated networks and data from mouse hippocampal neurons for which ground-truth connectivity data is available. Overall, I think this is a valuable application of these Fokker-Planck approaches to an inference problem, and suggests more principled ways of building integrate-and-fire models to match to data. The paper is well-written and clear, so I have only a few comments.

Comments:

1) In Section 2, why was the membrane time constant set to the true value? Could it not also be estimated?

We have revised Results section 2 (Inference of background inputs), taking explicitly into account the membrane time constant. In particular, we have included it in our estimations and provided a detailed explanation why it can be fixed at a reasonable value.

2) I would like to see an expanded discussion of the issue of non-stationarity (page 10) in the main text. This seems like it would be a major issue for any in vivo recording. Currently, the text mentions two methods and an analysis of variations of mean input at three different timescales, but the information provided is insufficient to understand what actually is being performed. The Methods section is more helpful, but could use additional justification of the methods (why three timescales, and what is the justification for the "combined timescale" estimate?).

We have elaborated on this issue in Results section 4 (Inference of synaptic coupling). The optimal timescale is a priori not clear, therefore we examined how this parameter affects detection accuracy. Interestingly, it influenced detection performance only weakly, which may be explained by the large signal-to-noise ratio in that dataset.

For the "combined timescales" estimates we aimed to optimize detection sensitivity by using the largest absolute z-score across the three timescales for each postsynaptic neuron. We have removed this part because that procedure did not yield improved results.

3) I was unclear why the input standard deviation in Fig. 7 wouldn't change between models with and without adaptation. Is there a reason that the two parameters should be completely decoupled from one another in the inference? This should be explained more fully.

We have clarified this point in Results section 5 (Inference of neuronal adaptation). Indeed, the presence of adaptation in the model affects the correlation coefficient between estimated and empirical input standard deviation only very weakly (see **Fig. S9**). This can be explained by the fact that the adaptation variable varies slowly (estimated adaptation time constant $> 4x$ membrane time constant on average); therefore, it affects the ISI mean more strongly than its variance which is predominantly influenced by the fast input fluctuations (parameter σ). Consequently, the estimated σ is much less affected than the estimated mean input (μ).

Reviewer #2 (Remarks to the Author):

The authors develop maximum likelihood based approaches to fitting integrate-and-fire models to extracellular recordings, specifically approaches that allow to use data that consists solely of spike times. Since population extracellular recordings are far more common than access to the membrane voltage of a population of neurons, there is value in such approaches.

My main concern is that the paper addresses a problem that has been repeatedly considered in prior work, yet is unclear about comparison to existing techniques. The paper considers multiple inference scenarios and I am still unclear as to in which (if any) of these scenarios does the approach allow biologically interesting information to be learned in a qualitatively superior way to that possible in previous approaches. I would also be interested if that is not the case, and the approach is just more computationally efficient, quantitatively more accurate, can be used with less data, or perhaps just more principled. But whichever of these claims are true they need to be clearly described and quantified for each scenario considered. As the paper stands I found this information too difficult to parse out and cannot recommend publication in its current format.

Major comments:

1. As stated above, my main concern is that I found it extremely difficult to parse out what specifically this approach enables that cannot be done with previous methods, or the specific advantages over existing methods. The paper considers multiple scenarios:

- i. Mean and standard deviation of input to a single neuron.
 - a. Synthetic data
 - b. In-vitro
- ii. EPSP shaped perturbations
 - a. Simulation fully observed
 - b. Simulation partially observed
- iii. Synaptic inference
 - a. Simulation fully observed
 - b. Simulation partially observed
- c. In-vivo
- iv. I&F models with adaptation
- v. Estimation of neuronal inputs from in-vivo recordings

The authors did include quantitative comparison to existing techniques in the results or figures. Some of these, for instance item i, have been directly addressed by previous approaches and many of these could have easily been compared to a variety of previous approaches, either more principled approaches or more straightforward approaches. The authors often report high accuracy for their results, but such numbers are very difficult to interpret as absolute numbers without reference to a comparison value. For each item the authors must either present the most pertinent comparison, or ideally comparisons to multiple existing techniques, while including straightforward extensions where necessary (e.g., grid searches for simple unknown but necessary parameters such as mean input current). Then the authors can clearly layout what specifically their approach improves: accuracy, computation time, amount of data necessary, etc. Where there is no substantial improvement, as I would expect to be the case for instance in item i, the authors can either present this as a useful introductory analysis, or claim that they prefer their technique since it is more principled, or something else. But this has to be much clearly and explicitly presented per result and not just in a somewhat vague Discussion paragraph.

We thank the reviewer for their extensive and constructive comments.

We have extended and reorganized the manuscript to include explicit comparisons with several alternative approaches across all inference scenarios (background inputs, input perturbations, synaptic coupling, and neuronal adaptation). We now compare our methods to 1) straightforward, model-free techniques based on cross-correlograms (CCGs), 2) established, model-based methods which rely on Poisson point processes and point process GLMs, and 3) a related approach based on I&F neurons (Pillow et al. 2005, Paninski et al. 2004):

- For inference of background inputs, input perturbations and neuronal adaptation (Results sections 2, 3, and 5) we benchmarked our approach on synthetic data against a related I&F-based method proposed by Pillow et al. (2005) both in terms of estimation accuracy, including detection sensitivity for input perturbations, and computation time (**see Figs. S1c,d, S3a-e and S8a-d**). An efficient implementation for that method is freely available, as was pointed out by Reviewer 3. We ensured equal conditions for the benchmarks in terms of model dynamics, parameters to be estimated, amount of observed data and

computational resources. Note that the descriptions of neuronal inputs and adaptation in the two I&F models differ. Note further that the method from Pillow et al. (2005) is not directly applicable to inference of synaptic coupling in networks (Results section 4). Across all benchmark tests our methods show clear improvements on estimation accuracy (and detection sensitivity) as well as a substantial reduction of computation times, up to orders of magnitude.

- In Results section 2 we further considered a Poisson point process for likelihood-based comparison on in-vitro and in-vivo data (**see Fig. 2d,g**). The I&F model is the preferred one according to the Akaike information criterion across all cells/stimuli/conditions.
- In Results sections 3 (input perturbations) and 4 (synaptic coupling) we included model-free, CCG-based methods for direct comparisons on detection accuracy (**see Figs. 3c,d, S3c, 4b-d, 5e-h, S5b-f, S6 and S7b,c**). Our methods outperformed the CCG approach in most comparisons, with only one exception where both approaches performed similarly (Fig. S7, see response to point 2 below) due to high signal-to-noise ratios.
- On inference of synaptic coupling in terms of detection and estimation of relative strengths of sub-sampled networks we additionally compared our approach to a method based on a point process GLM that is well constrained for the considered scenario (**see Figs. 5d,f,g,h, S5b-f and S6**). In particular, we chose a GLM that is tailored to capture the spiking dynamics in the synthetic data with minimal number of parameters. Our method performed clearly better on all tests.

Note that point process GLMs are flexible models which can be designed and optimized to well fit observed spike trains (see, e.g., Pillow et al. 2008, Stevenson et al. 2012). However, that approach is prone to overfitting unless strong constraints and/or regularization are enforced (see, e.g., Pillow et al. 2008, Stevenson et al. 2012, Aljadeff et al. 2016, Gerhard et al. 2017). An advantage of our approach in this respect is that the number of model parameters for optimization is comparably small, which strongly reduces the risk of overfitting, without sacrificing essential aspects of neural dynamics (such as refractoriness, adaptation or a resistive membrane). We have explained the relation to GLM-based methods in Results section 4 and in the Discussion.

On the in-vitro perturbation dataset another, yet indirect, comparison between our approach and a GLM method is possible by comparing Fig. 4c (method 2) with Fig. 2G in Volgushev et al. (2015). Detection time of our method 2 appears to be shorter overall.

Furthermore, we have improved our discussion on related I&F-based methods.

2. Related to this point, Figure 5 which discusses inference of synaptic connectivity on experimental data, appears to describe the results that are closest to what the paper aims to do, or at least that is what I understand inferring microcircuits, and the data described come with associated semi-ground truth. Therefore the lack of comparison here is most frustrating. The ROC curves look very good, which seems impressive, yet the empirical CCG presented shown in bottom left of Fig. 5A also looks very different from background, leading one to assume that straightforward techniques would also work quite well. As a matter of fact, reference 35, from which the authors took the data, also claims accurate inference of connections from CCG itself.

Perhaps the current approach does better, but explicit comparison has to be made to allow a reader to appreciate that.

We have included explicit comparisons with the model-free CCG approach (see Fig. S7). Indeed, the CCG method resulted in comparable detection accuracy, which may be explained by the large signal-to-noise ratio in this dataset, as English et al. (2017) focused on strong PYR-INT connections. This is now explained in Results section 4.2.

Note, however, that our approach yields higher detection sensitivity on in-vitro ground truth data (Results section 3.2) and improved accuracy on synthetic network data (Results section 4.1) compared to the CCG method when the signal-to-noise ratio is lower.

3. I couldn't understand the results related to subsampling observed neurons from a larger simulated network, which is the more relevant regime to look at for simulated data. The authors simply state that: "Estimation accuracy of connectivity structure and coupling strengths was surprisingly good for sufficiently long simulated recordings in this scenario." This is indeed surprising and requires more clarification.

We thank the reviewer for this comment. Inference of synaptic coupling from sub-sampled networks has now become a key part of the revised manuscript (Results section 4, see Figs. 5c-h, S5b-f and S6). We explained the challenges caused by shared input from unobserved neurons, the roles of connection density, delays, coupling strengths and measurement noise in this regard, and we clarified how those impede the different methods (our I&F-based vs. GLM-based and CCG-based).

A general challenge in sub-sampled networks arises from spike train correlations at small time lags generated by shared connections from unobserved neurons, regardless of whether a direct connection is present. These spurious correlations strongly impair our ability to distinguish the effects of synaptic connections from those caused by correlated inputs using CCGs, especially when coupling delays are small. Systematic comparisons between the three methods show that our I&F method masters this challenge best (Figs. 5c-h, S5b-f and S6).

One of the benefits of our approach is that it includes an explicit, principled mechanism to account for the effects of unobserved neurons, which are absorbed in the estimated statistics of the fluctuating background inputs. Correlated fast input fluctuations are not directly modeled, their effects are compensated for in the estimation of the background input parameters, whereas shared input dynamics on a longer timescale are explicitly captured by slow variations of the mean input for each neuron. This facilitates the isolation of pairwise synaptic interactions from common drive. Furthermore, we corrected for a potential bias in coupling strength estimates, due to effects of unobserved neurons that are not captured otherwise, by our inference method.

In sum, our approach yields accurate inference results for sub-sampled networks as long as the correlations between the hidden inputs, due to shared connections from unobserved neurons, are not too large. In particular, it outperforms classical CCG-based and GLM-based methods.

4. The logic of section 6 seemed strange to me. The main point seems to be Figure 8E-F, the balance of excitation and inhibition inferred by their model, but the authors go through a path of

claiming that their models are effective that is unclear and not entirely convincing to me. The authors assess their approach by comparing it to simpler models, but these models appear to be rather weak models: “To assess whether the complexity of this model is adequate given the available data we considered simpler models for comparison. For spontaneous activity, we compared our model to a Poisson process with constant rate. For click-evoked activity, we examined two additional models: (i) the leaky I&F model with only one (either excitatory or inhibitory) click-triggered input described by a delayed alpha function; (ii) a model in which the stimulus induced only a constant additional input.” Each of these comparisons takes out a key component that is very likely to be important and therefore neither of these comparisons seem very informative. It was difficult for me therefore to judge the reasonableness of Figure 8E-F. The authors make strong claims regarding this point: “Fitting only spike-trains, our approach therefore uncovers fundamental constraints on synaptic inputs and the local microcircuit without access to the values of the intra-cellular membrane potential.” Accordingly, more support and explanation would be helpful.

Given the substantial extensions of the manuscript performed during the revision, we have decided to remove the part on inference of click-evoked inputs from extracellular recordings in behaving ferrets, because these data do not involve ground truth information. An expanded analysis using our methods is planned for a separate study that is targeted towards auditory neuroscientists and will be presented in a more specialized contribution.

5. Though this comment is subjective, I found the paper’s tendency to oversell its results unappealing. Specifically, the paper pushes some of the main terminology beyond their natural usage. First, the authors refer to I&F models as mechanistic models. This is simply not the case, it is a phenomenological model. I refer the authors to the first paragraph of Abbott (1999) on the I&F model: “In 1907, long before the mechanisms responsible for the generation of neuronal action potentials were known, Lapique developed a neuron model that is still widely used today [3,7]. This remarkable achievement stresses that, in neural modeling, studies of function do not necessarily require an understanding of mechanism. Significant progress is possible if a phenomenon is adequately described, even if its biophysical basis cannot be modeled.”. Moreover multiple other papers referenced by the authors (if not all of them) properly refer to I&F models as phenomenological models. I&F models can certainly be viewed as closer to mechanism than abstract statistical models, as the authors claim, but this does not justify calling it a mechanistic model. Similarly, the paper claims to be “Inferring and validating mechanistic models of neural microcircuits based on spike-train data” but most of the results are inference of inputs to single neurons, which is not the standard usage for microcircuit models (though admittedly this is a not well defined term). It appears that the authors feel that single I&F models can be referred to as a microcircuit model: “We fitted a simple microcircuit model to these data. In the model, a leaky I&F model neuron received feed-forward excitatory and inhibitory inputs triggered by the clicks, as well as a fluctuating background input.” But this is hardly what comes to my mind when one refers to a microcircuit model. I understand the desire to make the paper more likely to be read by a general audience, but I feel the authors would be better served by not stretching definitions.

Following the reviewer's comment, we have reexamined the language used in the paper and avoided "mechanistic" as much as possible. In our revised manuscript, especially in the Introduction, we have clarified the position of I&F models with respect to more detailed models. We now present them as an intermediate level of description, more mechanistic than abstract statistical models (as the reviewer agrees) but less biophysically detailed than, e.g., models of the Hodgkin-Huxley type. Note that the understanding of I&F models has evolved in the last decade. Building upon the classical view of a neuron as a threshold device, these models have been advanced in recent years to account for the diverse electrophysiological features of neurons (Brunel et al. 2014): they effectively describe the dynamics of the neural membrane voltage and can be equipped with several mechanisms (for example, concerning spike initiation, adaptive excitability or distinct compartments; see, e.g., Fourcaud-Trocme et al. 2003, Ladenbauer et al. 2014, Ostojic et al. 2015) to generate diverse spiking behaviors and model multiple neuron types (see, e.g., Teeter et al. 2018). Note that the approximations in the I&F class compared to models of the Hodgkin-Huxley type are particularly warranted when quantitatively characterizing neural circuits based on spike-train data only, as multiple sets of biophysical parameters in those complex models can reproduce identical firing patterns (Prinz et al. 2003, Marder & Taylor 2011).

For the sake of compactness, we have kept "mechanistic" in the title, but we will remove it if the reviewer feels strongly against it.

Regarding the "microcircuits", note that we have removed the part on inference of click-evoked inputs (see previous response), hence "microcircuit" is solely used within the context of inferring connectivity.

Minor comments:

1. In general I found multiple figures to be crowded and ineffective. There is often a lot of detail in the figure that is very difficult to know how to interpret while crucial detail remains difficult to judge. For instance, Figure 8A bottom appears to contain crucial information regarding the comparison that will ultimately be made in Figure 8E-F, but it is tiny, not well-described and overwhelmed by the much larger (and seemingly less informative) subplots 8B-D.

We have completely revised all figures, particularly taking into account the reviewer's suggestion. Note that the previous Fig. 8 has been removed (see response above).

2. Often important information regarding the results is present only in the methods, and sometimes sparsely described there, for instance the sensitivity measure used in Figure 3D.

We have included important methodological information in the Results and improved the Methods sections.

3. Percentile is sometimes written as "%-ile". This is non-standard usage and should be avoided.

We have improved the notation as suggested.

4. Section regarding figure 4: “The estimation accuracy of synaptic strengths is still good in case of weak correlations of the external input fluctuations (correlation coefficient $c \leq 0.05$ for each pair of observed neurons) but clearly decreases as these correlations increase (Fig. 4E). In particular, positive correlations $c > 0$ lead to an overestimation of the coupling strengths.” Figure 4E shows only two scatters, one for $c=0.05$ and one for $c=0.1$. The Pearson’s correlation of the former is 0.95 and the latter 0.94. The described shift up of weights can be observed but is not quantified, and the quantification doesn’t show a decrease in accuracy, which makes relating the figure to the text confusing.

We have included an additional measure, the mean absolute error, to better quantify any bias in the estimated coupling strengths (cf. Figs. 5b,d, S5b-f and S6). Note that the entire Results section 4 has been substantially revised and extended.

5. Top of page 11: “for the combined timescales variant; when using all instead of max. thresh 1000 PYR spikes...”. Max. as an abbreviation should be avoided in the text.

We have adjusted the notation as suggested.

6. Page 11 describing adaptation: “This behavior is typically mediated by a slowly decaying transmembrane potassium current, which rapidly accumulates when the neuron spikes repeatedly.” I couldn’t tell with certainty if this is meant as an abbreviated description of calcium dependent potassium currents or not. The authors don’t have to mention this, but if they do a clearer description (including a reference) would be useful.

We have improved the description, including two suitable references.

7. As stated above I wouldn’t call section 6 fitting a simple microcircuit model.

This section has been removed, as explained above.

8. The authors sometimes include words in parentheses for unclear reasons. For instance: “This enables to better isolate pairwise interactions from (ubiquitous) fluctuating drive...” either ubiquitous is an important word in this center and should be retained without parentheses or it can be removed. Unclear usage of parentheses can be confusing.

We have revised our usage of parentheses throughout, and removed them at places where ambiguous interpretation could not be excluded.

9. “Neurons were separated into PYRs and INTs according to their spiking statistics.” The term spiking statistics could be read to indicate that spike waveform was not used, which is not the case, at least in the paper that the authors reference.

We have clarified this point.

Reviewer #3 (Remarks to the Author):

In this manuscript, Ladenbauer and colleagues present a probabilistic approach to fit integrate and fire models to extracellularly recorded neurons. The methods are used to infer connectivity between neurons and stimulus-response relationships. This method aims to bridge the gap between purely phenomenological tools and biophysical descriptions of neurons.

Unfortunately, I have strong concerns about the novelty of the methods and the strength of the claims that the LIF approach improves upon purely statistical models. Thus, while the general goals behind this methods paper are of wide interest, I cannot recommend this paper to be published in Nature Communications at this time.

We thank the reviewer for the constructive comments. In response, we have substantially extended our study, by 1) including systematic comparisons with related methods, 2) expanding and advancing our results on inference of synaptic couplings in sub-sampled networks, 3) incorporating an additional ground truth dataset. Our revised manuscript now includes validations using ground truth data from in-vitro and in-vivo recordings for all inference scenarios, and benchmarks against several existing methods.

We would further like to highlight the novelty of our methods for the estimation of couplings in spiking network models in an in-vivo setting; related previous approaches have been designed for single I&F neurons (Pillow et al. 2005, Paninski et al. 2004) or networks limited to vanishing input fluctuations (Cocco et al. 2009).

My main critique of this paper concerns the novelty of the integrate-and-fire methodology.

Several of the modeling problems presented in the results section fall within the generalized integrate and fire model from Pillow et al. (2005), for which code is publicly available.

Comparing the results from Ladenbauer and colleagues to equation 1 from Pillow et al:

- results section 2 considered the case where $I_{\text{stim}0}=\text{constant}, I_{\text{spk}}=0$.
- results section 3 uses the model with $I_{\text{spk}} = 0$ the stimulus is μ_{1}
- results section 5 includes the I_{spk} term which captures adaptation
- results section 6 applies the full generalized IF model to spiking data in response to a sensory stimulus, as Pillow et al did with retinal ganglion cells (but with a specific decomposition of the linear stimulus filter).

Given previous work by Paninski and colleagues cited in this manuscript that established the log concavity of the generalized IF's likelihood function, accurately recovering the IF model's parameters with maximum likelihood estimation given enough data in the simulation studies here was expected.

We thank the reviewer for pointing out this study and the available code. In the revised manuscript, we benchmarked our methods against the related approach from Pillow et al. (2005) using synthetic data in all applicable scenarios: inference of background inputs, input perturbations and neuronal adaptation (Results sections 2, 3, and 5). We compared estimation accuracy, detection sensitivity (for input perturbations), and computation time (**see Figs. S1c,d, S3a-e and S8a-d**). We ensured equal conditions for the benchmarks in terms of model dynamics, parameters to be estimated, amount of observed data and computational resources. Note that the method from Pillow et al. (2005) is not directly applicable to inference of synaptic coupling in networks (Results section 4). Across all benchmark tests our methods show clear

improvements on estimation accuracy and detection sensitivity as well as a substantial reduction of computation times, up to orders of magnitude.

One of the major contributions of this paper is estimating connectivity between extracellularly recorded neurons. The discussion states that the IF model can “better isolate pairwise interactions from (ubiquitous) fluctuating drive”. The introduction claims that statistical models suffer from a lack of connection to biophysical structures and that this paper serves to bridge the gap between biophysical and statistical models. However, the authors have not cited an extremely similar study using generalized linear models (GLMs) to tackle the problems in results sections 3-4 (while also including adaptation): Volgushev et al. (2015). Thus, these claims are not substantiated because the LIF model isn't compared to anything like GLMs to show that the biophysical approach shows more than can be seen with a phenomenological model.

We have applied our methods to the in-vitro ground truth data from Volgushev et al. (2015) and directly compared them to a CCG-based method in terms of detection performance (**see Fig. 4**). These results validate our approach and demonstrate that it is more sensitive than the model-free CCG method. Furthermore, it appears more sensitive compared to the GLM method used in Volgushev et al. (2015): detection times of our method 2 appear to be shorter overall by comparing Fig. 4c with Fig. 2G in Volgushev et al. (2015). We focused on the first experiment in Volgushev et al. (2015), which involves one artificial presynaptic spike train for each recorded neuron (the second experiment in that paper considered multiple presynaptic spike trains per neuron, which were independently generated).

The data from Volgushev et al. (2015) constitute a valuable testbed for methods that aim to infer synaptic couplings. However, these data do not include some of the challenging obstacles caused by unobserved neurons in vivo, such as input correlations in a partially observed network due to unobserved neurons, and nonstationary drive. We focus on these issues in Results section 4: inference of synaptic coupling from sub-sampled networks. We have significantly expanded the corresponding results, and this section has become a key part of the revised manuscript (**see Figs. 5c-h, S5b-f and S6**). We compare our approach to CCG-based and GLM-based methods, explain the challenges caused by shared input from unobserved neurons and the roles of connection density, delays, coupling strengths and measurement noise in this regard.

A general challenge in sub-sampled networks arises from spike train correlations at small time lags generated by shared connections from unobserved neurons, regardless of whether a direct connection is present. These spurious correlations strongly impair our ability to distinguish the effects of synaptic connections from those caused by correlated inputs using CCGs, especially when coupling delays are small. Systematic comparisons between the three methods show that our I&F method masters this challenge best. One of the benefits of our approach is that it includes an explicit, principled mechanism to account for the effects of unobserved neurons, which are absorbed in the estimated statistics of the fluctuating background inputs. Correlated fast input fluctuations are not directly modeled, their effects are compensated for in the estimation of the background input parameters, whereas shared input dynamics on a longer timescale are explicitly captured by slow variations of the mean input for each neuron. This facilitates the isolation of pairwise synaptic interactions from common drive. Furthermore, we

corrected for a potential bias in coupling strength estimates, due to effects of unobserved neurons that are not captured otherwise, by our inference method.

In sum, our approach yields accurate inference results for sub-sampled networks as long as the correlations between the hidden inputs, due to shared connections from unobserved neurons, are not too large. In particular, it outperforms classical CCG-based and GLM-based methods.

I would still encourage the authors to build on the novel content of this paper for future publications. This includes:

1. Faster estimation of the stochastic LIF model. How do methods 1a,b and 2 compare in speed and accuracy to previous Fokker-Planck estimation tools? The code from Pillow et al is available from the Pillow lab website for comparison.

We have included detailed benchmarks against the method from Pillow et al. (2005), using the efficient, available implementation (based on Matlab with integrated C code for accelerated program execution) in all applicable scenarios. Our methods show clear improvements on estimation accuracy, detection sensitivity, and computation time across all tests (**see Figs. S1c,d, S3a-e and S8a-d**).

2. Application of the generalized LIF to ferret cortex. What do we learn from this model fit?

We have removed the part on inference of click-evoked inputs from extracellular recordings in behaving ferrets (former Results section 6), because these data do not involve ground truth information. An expanded analysis using our methods is planned for a contribution on task-related activity in auditory cortex which goes beyond the scope of this paper. Here, we focused on the inference methodology and, in particular, their validation using synthetic data and various ground truth recordings.

3. LIF for estimating connectivity. How does this method compare to the GLM estimator in Volgushev et al? Does this biophysical tool validate the more abstract GLM approach? Or does the biophysical model pick up on things the GLM misses? Data, but not code, from the Volgushev paper is available from the Stevenson lab website.

We validated our methods using the in-vitro ground truth data from Volgushev et al. (2015) and directly compared them to a CCG-based method in terms of detection performance (**see Fig. 4**). The results demonstrate that our approach is more sensitive than the model-free CCG method. Furthermore, it appears more sensitive compared to the GLM method used in Volgushev et al. (2015): detection times of our method 2 appear to be shorter overall by comparing Fig. 4c with Fig. 2G in Volgushev et al. (2015).

Note that on inference of synaptic coupling in sub-sampled networks we additionally compared our approach to a method based on a point process GLM (**see Figs. 5d,f,g,h, S5b-f and S6**). In particular, we chose a GLM that is tailored to capture the spiking dynamics in the synthetic data with minimal number of parameters. Our method performed clearly better on all tests.

References:

Pillow JW, Paninski L, Uzzell VJ, Simoncelli EP, Chichilnisky EJ. (2005). Prediction and Decoding of Retinal Ganglion Cell Responses with a Probabilistic Spiking Model. *Journal of Neuroscience* 25:11003-11013.

Volgushev M, Ilin V, Stevenson IH (2015). Identifying and Tracking Simulated Synaptic Inputs from Neuronal Firing: Insights from In Vitro Experiments. *PLoS Comput Biol* 11(3): e1004167.doi:10.1371/journal.pcbi.1004167

We thank the reviewer for the suggestions and for pointing to these relevant references.

Specific comments and suggestions:

- The presentation of statistical analyses was difficult to follow. Population and perturbation results (Figs 3-5) use the likelihood formulation for fitting the IF models, but do not appear to take advantage of the probabilistic formulation for model selection/hypothesis testing (does a connection exist? Is $J > 0$?). For example, I found the methods used to calculate detection sensitivity in Figure 3D unclear and difficult to compare between CCG and LIF modeling. It was notable for the adaptation/in vivo results (results sections 5 & 6) changed to use AIC for model selection; the paper would benefit from applying this approach more consistently.

We now use the AIC model comparison in Results section 2 (background input), where we compare I&F and Poisson models, and in Results section 5 (neuronal adaptation), where we compare adaptive and nonadaptive I&F models. We included a likelihood-based technique for detecting input perturbations in Results section 3.2 (cross-validated log-likelihood ratio, adopted from Volgushev et al. 2015) for comparison, in addition to techniques based on surrogate data where the effects of perturbations are absent (by simulations in case of synthetic data) or washed out (by perturbing spike times). In this way, the two different methods, I&F-based and model-free, CCG-based, can be compared by the same technique to detect significance (using surrogate data, since likelihood-based techniques cannot be applied to the CCG method).

- Equations 15 & 50 show that the coupling inputs are limited to delta functions (in contrast to Volgushev et al). Bottom left of Fig 5A suggests that the delta shape misses prediction of the true CCG. Does this approximation limit/bias detection of more realistic EPSP shapes?

We tested this point explicitly on the ground truth data from Volgushev et al. (2015) where artificial PSCs had a rise time of 1 ms and a decay time of 10 ms. In Results section 3.2 we used both alpha functions or delayed delta pulses to model PSCs (with two parameters each). Overall, both kernels lead to accurate results in detecting weak input perturbations, compared to the model-free CCG method.

- Figure 4: C-D – Estimated inhibitory coupling weights appear biased towards 0. Are excitatory weights estimated more accurately?

E - Is the effect of correlation a constant positive bias? How does this compare with shuffling-based CCG methods for detecting pairwise correlations?

We have significantly expanded Results section 4.1, where we focus on partially observed networks and compare our approach to CCG-based and GLM-based methods; former Fig. 4 has been replaced by current Figs. 5, S5 and S6. We have introduced a simple procedure to correct for a potential systematic bias in the estimated coupling strengths (see Methods section 7.1). The magnitudes of both excitatory and inhibitory weights are sometimes underestimated by our method (see, e.g., Fig. S5e,f) and overall, inhibition appears to be slightly more affected. This asymmetry is also noticeable for the other two methods, it can be intuitively understood from the CCG method: detecting weak inhibitory connections is particularly challenging because of increased correlations for small time lags due to unobserved neurons (see, e.g., Fig. 5h(ii)). This is now explained in the Results text.

- In section 6, the excitatory and inhibitory components of the stimulus response are additive. Is this decomposition of input into these components unique? And can they be compared to recorded synaptic currents (if available)?

We have removed this section, as explained above.

- Section 4.2 and Fig 5A – The method for estimating the mean rate appears ad hoc, and is not justified in simulation. Page 10 indicates that 3 timescales were used to estimate the mean rate and the results were combined “using the largest absolute z-score across the three timescales for each connection.” I am unsure what this means.

In Results section 4.2 we account for nonstationarity in a straightforward and simple way, which involves an additional parameter, the width of the Gaussian kernel that determines the timescale of the mean input variations. Indeed, the optimal value is a priori not clear. Our results show that this timescale influences detection performance only weakly, which may be explained by the large signal-to-noise ratio in the dataset.

For the “combined timescales” estimates we aimed to optimize detection sensitivity by using the largest absolute z-score across the three timescales for each postsynaptic neuron. We have removed this part because that procedure did not yield improved results.

References

Aljadeff J, Lansdell BJ, Fairhall AL, Kleinfeld D. *Analysis of Neuronal Spike Trains, Deconstructed*. Neuron 2016

Brunel N, Hakim V, Richardson MJE. *Single neuron dynamics and computation*. Curr. Opin. Neurobiol. 2014

Cocco S, Leibler S, Monasson R. *Neuronal couplings between retinal ganglion cells inferred by efficient inverse statistical physics methods*. Proc. Natl. Acad. Sci. USA 2009

English DF, McKenzie S, Evans T, Kim K, Yoon E, Buzsaki G. *Pyramidal Cell-Interneuron Circuit Architecture and Dynamics in Hippocampal Networks*. Neuron 2017

Fourcaud-Trocmé N, Hansel D, van Vreeswijk C, Brunel N. *How spike generation mechanisms determine the neuronal response to fluctuating inputs*. J. Neurosci. 2003

Gerhard F, Deger M, Truccolo W. *On the stability and dynamics of stochastic spiking neuron models: Nonlinear Hawkes process and point process GLMs*. PLOS Comput Biol. 2017

Ladenbauer J, Augustin M, Obermayer K. *How adaptation currents change threshold, gain and variability of neuronal spiking*. J. Neurophysiol. 2014

Marder E, Taylor AL. *Multiple models to capture the variability in biological neurons and networks*. Nat. Neurosci. 2011

Ostojic S, Szapiro G, Schwartz E, Barbour B, Brunel N, Hakim V. *Neuronal Morphology Generates High-Frequency Firing Resonance*. J. Neurosci. 2015

Paninski L, Pillow JW, Simoncelli EP. *Maximum Likelihood Estimation of a Stochastic Integrate-and-Fire Neural Encoding Model*. Neural Comput. 2004

Pillow JW, Paninski L, Uzzell VJ, Simoncelli EP, Chichilnisky EJ. *Prediction and Decoding of Retinal Ganglion Cell Responses with a Probabilistic Spiking Model*. J. Neurosci. 2005

Pillow JW, Shlens J, Paninski L, Sher A, Litke AM, Chichilnisky EJ, et al. *Spatio-temporal correlations and visual signalling in a complete neuronal population*. Nature 2008

Prinz AA, Billimoria CP, Marder E. *Alternative to Hand-Tuning Conductance-Based Models: Construction and Analysis of Databases of Model Neurons*. J. Neurophysiol. 2003

Stevenson IH, London BM, Oby ER, Sachs NA, Reimer J, Englitz B, et al. *Functional Connectivity and Tuning Curves in Populations of Simultaneously Recorded Neurons*. PLOS Comput. Biol. 2012

Teeter C, Iyer R, Menon V, Gouwens N, Feng D, Berg J, et al. *Generalized leaky integrate-and-fire models classify multiple neuron types*. Nat. Commun. 2018

Volgushev M, Ilin V, Stevenson IH. *Identifying and Tracking Simulated Synaptic Inputs from Neuronal Firing: Insights from In Vitro Experiments*. PLOS Comput. Biol. 2015

Reviewers' Comments:

Reviewer #1:

Remarks to the Author:

The authors have made a number of changes that have improved the manuscript. I am now satisfied to recommend the manuscript for publication.

Reviewer #2:

Remarks to the Author:

The authors have improved the manuscript, adding much needed comparisons to previous literature and different approaches.

Regarding non-synthetic data, performing direct comparison to the straightforward CCG method the authors show some improvement in one dataset (Figure 4) and comparable results in another (Figure S7).

Reviewer #3:

Remarks to the Author:

The authors have provided extensive revisions that address many of my previous concerns about the scope of this paper within the existing literature on fitting stochastic I&F models to spike trains. The time and effort spent reframing and expanding this work has resulted in a greatly improved manuscript. I only have a few minor comments.

I still have a few remaining comments about the comparison to GLM methods:

1. The exact methods in Figure 4 do not appear to correspond exactly to the methods used in Volgushev et al Figure 2. Therefore, it's hard to compare the performance quantitatively. This would be improved by providing a GLM fit with the same 5-fold cross-validation and LL ratio comparison and adding the results as a column in Figs 4A,C.
2. For the network GLM in section 4, could you provide more detail on the parametrization of the coupling terms so that it's easier to compare to the I&F model without needing to dig through Zaistevev et al? This is necessary to judge whether the I&F estimation is due to the mechanistic model or differences in particular parametrization used to describe coupling in the two methods.
3. Last paragraph of section 4.2 says the GLM method had greatly increased computational cost. How is this quantified?
4. My point here is opinionated, but something for the authors to consider. Paragraph bottom of page 15 and top of 16: GLM "is prone to overfitting unless strong constraints or regularization are enforced." I'd argue that the I&F methods here constitute such "strong constraints" on the assumptions in the GLM. This is by no means a negative aspect, but I suggest framing this argument in favor of the mechanistic approach (biophysically guided constraints, rather than more abstract constraints) as a more accurate characterization.
5. This point is only a suggestion (and may not currently have an answer): It'd be nice to see this model framed within existing literature connecting GLMs and stochastic IF models (see Mensi, Naud, & Gerstner 2011 "From stochastic nonlinear integrate-and-fire to generalized linear models"). Is there

an intuitive reason why the more biophysical formulation gives better coupling estimates than the GLM?

Response to Reviewer 3

We thank the reviewer for their supportive comments. We have now implemented all the suggested changes. Below we provide a detailed, point-by-point response (comments in blue, answers in black).

1. The exact methods in Figure 4 do not appear to correspond exactly to the methods used in Volgushev et al Figure 2. Therefore, it's hard to compare the performance quantitatively. This would be improved by providing a GLM fit with the same 5-fold cross-validation and LL ratio comparison and adding the results as a column in Figs 4A,C.

To make our Figure 4 directly comparable to the Figure 2 of Volgushev et al., we have re-run our analyses with 10-fold cross-validation as in Vogushev et al., and updated the figure. Our Fig 4c is now directly comparable with their Fig 2G, and our Fig 4a is directly comparable with their Fig 2F. We now indicate this explicitly in the text. Both comparisons show clear improvements of our methods with respect to theirs.

We would like to remark that 10-fold cross-validation lead to increased variance of test LLR across folds especially for short recording durations, which required to increase the number of short duration segments to obtain robust averages (that number of repetitions is not given in Volgushev et al.). These additional segments were also used in the CCG analysis (for reasons of comparability) which explains the small differences of revised Fig 4b and 4c (right) compared to the previous version.

Note that the GLM code used by Volgushev et al. is not publicly available, hence some details of the analysis may still differ (such as number of repetitions used). Since the figures are otherwise directly comparable between the two papers, fitting a different GLM to their data could be considered misleading, so we opted against it.

2. For the network GLM in section 4, could you provide more detail on the parametrization of the coupling terms so that it's easier to compare to the I&F model without needing to dig through Zaystev et al? This is necessary to judge whether the I&F estimation is due to the mechanistic model or differences in particular parametrization used to describe coupling in the two methods.

Coupling terms are equivalent in the two models. The Zaytsev et al. GLM model uses delayed delta pulses on the variable that is interpret as membrane potential (which is passed through the exponential function to produce a Poisson spike rate). We have now clarified this in the Methods.

3. Last paragraph of section 4.2 says the GLM method had greatly increased computational cost. How is this quantified?

This sentence does not refer to our analyses, but to work previously performed in the English, McKenzie et al. 2017 paper, where the data was originally published. Each GLM fit took between 200 - 400s on a GPU (depending on the length of the recording) versus 20ms for the simple CCG analysis, so the GLM analysis was at least 10,000 times slower. We now specify this in Results section 4.2.

4. My point here is opinionated, but something for the authors to consider. Paragraph bottom of page 15 and top of 16: GLM "is prone to overfitting unless strong constraints or regularization are enforced." I'd argue that the I&F methods here constitute such "strong

constraints" on the assumptions in the GLM. This is by no means a negative aspect, but I suggest framing this argument in favor of the mechanistic approach (biophysically guided constraints, rather than more abstract constraints) as a more accurate characterization.

We thank the reviewer for this suggestion, we have now edited the text at the top of p.16 to include it:

"An advantage of our approach in this respect is that the basic mechanistic principles included in I&F models provide a natural regularization and reduce the number of model parameters, which strongly reduces the risk of overfitting. Note that our method 2 is essentially based on mapping I&F models to simplified, constrained GLM-like models."

5. This point is a only a suggestion (and may not currently have an answer): It'd be nice to see this model framed within existing literature connecting GLMs and stochastic IF models (see Mensi, Naud, & Gerstner 2011 "From stochastic nonlinear integrate-and-fire to generalized linear models"). Is there an intuitive reason why the more biophysical formulation gives better coupling estimates than the GLM?

We believe that the reviewer refers here to the GLM model of Zaytsev et al. This model was not directly derived from an I&F model, so the connection here is not direct. Not that in contrast, our method 2 is directly related to mapping I&F models onto GLM-like models, as previously done in Ostojic & Brunel 2011 and Augustin, Ladenbauer et al. 2017. We have now clarified this in the Discussion paragraph that compares I&F models and GLMs (top of p.16).

Reviewers' Comments:

Reviewer #3:

Remarks to the Author:

The authors' response and revised manuscript have thoroughly addressed the few minor comments I had in the last round.

The authors have done an excellent job demonstrating feasible methods for fitting noisy IF neurons to spike train data, and I strongly recommend this paper for publication.